# Enhancing Sustainability through Accessible Health Platforms: A Scoping Review

Domenica Ramírez-Saltos [1], Patricia Acosta-Vargas [1,2,*], Gloria Acosta-Vargas [3], Marco Santórum [4], Mayra Carrion-Toro [4], Manuel Ayala-Chauvin [5], Esteban Ortiz-Prado [6], Verónica Maldonado-Garcés [7] and Mario González-Rodríguez [1]

1 Intelligent and Interactive Systems Laboratory, Universidad de Las Américas, Quito 170125, Ecuador; domenica.ramirez@udla.edu.ec (D.R.-S.); mario.gonzalez.rodriguez@udla.edu.ec (M.G.-R.)
2 Carrera de Ingeniería Industrial, Facultad de Ingeniería y Ciencias Aplicadas, Universidad de Las Américas, Quito 170125, Ecuador
3 Facultad de Medicina, Pontificia Universidad Católica del Ecuador, Quito 170143, Ecuador; gfacosta@puce.edu.ec
4 Departamento de Informática y Ciencias de la Computación, Escuela Politécnica Nacional, Quito 170525, Ecuador; marco.santorum@epn.edu.ec (M.S.); mayra.carrion@epn.edu.ec (M.C.-T.)
5 Centro de Investigación en Ciencias Humanas y de la Educación, Universidad Tecnológica Indoamérica, Ambato 180103, Ecuador; mayala5@indoamerica.edu.ec
6 One Health Research Group, Universidad de Las Américas, Quito 170125, Ecuador; esteban.ortiz.prado@udla.edu.ec
7 Facultad de Psicología, Pontificia Universidad Católica del Ecuador, Quito 170143, Ecuador; vmaldonado794@puce.edu.ec
* Correspondence: patricia.acosta@udla.edu.ec

**Abstract:** The digital transformation of healthcare platforms has ushered in a new era of accessibility, making health information and services widely available. This comprehensive scoping review delves into the accessibility landscape of health platforms by analyzing 29 carefully selected research articles. These studies employ automated tools and manual evaluations to evaluate platform accessibility comprehensively. This study revealed that (52%) of these articles are based on automated methods, while 34% combine automated and manual approaches. Most studies show compliance with the latest versions of the Web Content Accessibility Guidelines (WCAG), with a significant focus (70%) on compliance with level A. This study reveals recurring issues within the perceivable operable, understandable, and robust categories, underscoring the pressing need for strict the accessibility testing of health platforms. This study demonstrates substantial agreement between raters, reinforced by a Cohen's kappa coefficient of 0.613, indicating their reliability in classifying the articles. Future efforts should persist in refining accessibility standards, advocating for compliance with the WCAG, exploring innovative methods to assess the accessibility of healthcare platforms, and conducting user-centered research. This review highlights the paramount importance of ensuring equitable access to health information and services for people, regardless of their abilities or conditions, which resonates significantly with the issue of sustainability in healthcare and its socioeconomic and environmental implications.

**Keywords:** accessible; digital platforms; respiratory rehabilitation; scoping review; therapeutic education; sustainability



## 1. Introduction

In the current digital era, where the search for sustainability is paramount, the link between accessibility and health platforms plays a fundamental role. Online health platforms serve as gateways to crucial health information and services and have become instrumental in the search for sustainable healthcare solutions.

A health platform [1] is an IT system that provides a common infrastructure for delivering healthcare services. Organizations, including hospitals, clinics, home care providers, and governments, can use healthcare platforms.

Examples of healthcare platforms include electronic health records (EHRs) and computer systems that store patients' medical information, including health records, test results, and medications.

Laboratory information management systems (LIMS) are computer systems that automate laboratory testing. Medical imaging information management systems (RIS) are computer systems that store medical images, such as X-rays and MRIs. Patient care management systems (PACS) are computer systems that allow physicians to view medical images from anywhere. Patient health management (PHR) platforms are computer systems that allow patients to store and share medical information.

Healthcare platforms rapidly evolve to incorporate new technologies, such as artificial intelligence and machine learning.

Ensuring the accessibility of these platforms is not only an ethical imperative but also a fundamental component of a sustainable healthcare ecosystem. It is not simply about including all individuals but rather establishing an equitable and inclusive framework for healthcare that aligns with the core principles of sustainability.

This study recognizes that sustainability is not limited solely to ecological considerations but extends to social and ethical dimensions. Providing equitable and inclusive healthcare through accessible platforms is essential in this broader sustainability paradigm. The present research focuses on the accessibility of health platforms, examining their methodologies, tools, disabilities considered, and compliance with international accessibility guidelines.

Web accessibility improves sustainability by aligning with the Sustainable Development Goals (SDGs) [2]. SDG 4 relates to quality education by ensuring everyone has equal access regardless of their abilities. SDG 8 focuses on decent work and economic growth by opening up new opportunities for people with disabilities to participate in the workforce and contribute to the economy.

SDG 10, which reduces inequalities, ensures that everyone has equal access to information and services. SDG 16 helps to promote peace, justice, and strong institutions by ensuring everyone has equal access to government services and information.

In addition to these SDGs, web accessibility contributes to sustainable development's overarching goal by promoting inclusion and participation. We can create a more sustainable and equitable world when everyone has equal access to the web.

An accessible health platform not only considers the needs of those with visual or hearing disabilities but also those with cognitive, motor, and other disabilities. The World Wide Web Consortium [3] (W3C) has established guidelines known as the Web Content Accessibility Guidelines (WCAG) [4] to help developers and designers create accessible websites and platforms. However, the application of these guidelines in the context of health platforms and the effectiveness of the assessment tools are still under debate.

In this increasingly digital era, health platforms have risen to prominence and serve as critical conduits to deliver vital health information [5], resources, and services to people worldwide. As digital platforms evolve [6], so does the pressing need to ensure that they are universally accessible and serve a wide range of user capabilities. This research is not simply a matter of inclusion: it is about ensuring that health, a fundamental human right, remains equitable and available to all. The issue at hand revolves around the current state of accessibility of health platforms. Is the digital transformation of health leaving a significant part of the population behind, or are platforms adapting to become more inclusive?

The objective of this study was to provide a comprehensive overview of the landscape of the accessibility of healthcare platforms [6,7], clarifying the methodologies used to assess accessibility, the tools used, the extent of the disabilities considered, and compliance with international accessibility guidelines. Through this, we seek to provide insight into the

gaps and opportunities in the field and guide future efforts to make health platforms more universally accessible.

To gain a comprehensive understanding of the accessibility of health platforms, we conducted a scoping review of 29 selected articles. These articles were meticulously chosen based on their relevance to the accessibility of health platforms, encompassing both automated and manual evaluation methods.

Our review highlighted the importance of a two-pronged approach to assessing the accessibility of health platforms. Of the 29 articles, 12 incorporated real users and expert validation, emphasizing the human element in accessibility evaluation. Interestingly, eight of these articles used automated tools and manual reviews, suggesting a growing recognition of the need for a comprehensive evaluation methodology. This research is particularly crucial given that automated tools, while efficient, can sometimes produce inaccurate or incomplete results, requiring human validation.

Adherence to the WCAG emerged as a recurring theme, with several articles updating their compliance based on the latest versions of the WCAG. This research highlighted the field's dynamic nature and the importance of continuous updates to meet international accessibility standards. Notably, in some cases, there was a delay in adopting the latest versions of the WCAG [4], pointing to potential areas of improvement to keep health platforms up to date with the latest accessibility guidelines.

Ensuring the accessibility of health platforms is not a one-time task but rather an ongoing journey. Our review highlights the value of combining automated tools with manual reviews, offering a more holistic and complete assessment of platform accessibility. While many health platforms and related items have made commendable progress in adhering to the WCAG, there remains a pressing need for continued training, updates, and awareness. Only through sustained efforts can we ensure that health platforms are genuinely inclusive and leave no one behind in the digital health revolution.

This scoping study aims to explore the existing literature related to the accessibility of health platforms. It examines the tools used to assess accessibility, the disabilities considered in the assessments, the WCAG versions and conformance levels applied, and the results obtained. With this review, we seek to provide a comprehensive view of the current state of accessibility in health platforms, identify areas for improvement, and offer recommendations for future research and development in the field.

Accessibility in healthcare platforms is essential to address issues of equity in healthcare, reduce health inequalities, and promote social inclusion. These are vital aspects of the sustainability goals in the healthcare sector, which seek to ensure everyone has access to quality and equitable healthcare while optimizing resources and complying with ethical and legal regulations.

This research follows the following structure: Section 2 introduces the literature review on web accessibility and sustainability; Section 3 introduces the PRISMA method of the scoping review; Section 4 applies the review and presents the results, which are divided into two parts, a bibliometric analysis and a scoping review; Section 5 is dedicated to discussing the obtained results; finally, Section 6 addresses the conclusions derived from this research, its limitations, and directions for future work.

## 2. Literature Review: Web Accessibility and Sustainability

In recent years, the intersection between web accessibility and sustainability has gained significant attention within the research landscape [8]. The relationship between these two seemingly distinct fields is becoming increasingly evident, reflecting the broader paradigm shift toward sustainability in the digital age.

Accessibility and sustainability play a crucial role in the context of healthcare platforms [9]. Accessibility refers to the ability of these platforms to be used by all users, including those with disabilities [10]. On the other hand, sustainability relates to the ability of these platforms [11] to operate efficiently and effectively without harming the environment or society.

Ensuring the accessibility of healthcare platforms means that they must be easy to use and understand for everyone, regardless of whether they have visual, hearing, motor, or cognitive disabilities. There are several measures that platforms can implement, such as the inclusion of alternative text for images, audio descriptions in videos, the use of assistive technologies, and the adoption of intuitive and user-friendly user interface designs [4].

On the other hand, sustainability in healthcare platforms refers to operating efficiently and effectively without compromising the environment or society [12]. This involves the adoption of energy-saving technologies, the digitization of processes to minimize the use of paper, and the consideration of recycled materials for its operations.

Accessibility and sustainability present challenges for healthcare platforms, as implementing accessibility measures can be costly and complex, and some healthcare decision-makers may not be fully aware of the benefits of sustainability. However, despite these challenges, accessibility and sustainability are essential aspects that must be considered when designing and implementing healthcare platforms [8].

Assistive technologies, such as screen readers and virtual keyboards, are crucial for disabled people to access healthcare platforms [13]. The user-centered approach enables healthcare organizations to create accessible platforms for all users, including those with disabilities. In addition, innovative solutions are being explored, such as using virtual reality in hospitals to help patients with disabilities better understand their treatments.

In terms of sustainability, healthcare organizations are adopting energy-saving technologies, such as energy-efficient servers and efficient cooling systems, to reduce their energy footprint [2,12]. They are also implementing digital processes to minimize the use of paper and other resources. Investments in sustainability are aimed at reducing the environmental impact of these organizations.

These trends indicate that accessibility and sustainability [2] will remain crucial in developing healthcare platforms. As technologies and practices evolve, healthcare organizations will have new opportunities to improve both the accessibility and sustainability of their platforms.

Assistive technologies [14] are advancing rapidly, and recognizing more and more types of content, such as videos and images. On the other hand, healthcare organizations continue to adopt sustainable technologies and practices, such as using solar energy to power their facilities.

These trends are critical to ensuring that healthcare platforms are accessible and sustainable for all users.

Our review looks at the specific tools and methodologies mentioned in these articles. For automated evaluations, tools such as Achecker [15], WAVE [16], TAW [17], and others, stood out [18]. On the other hand, manual reviews covered methods such as validation based on user tasks, expert consultations, user feedback, and questionnaires. We delved deeper into articles that used automated tools and manual reviews, emphasizing the synergies and discrepancies between the two evaluation methods. The articles were also examined to understand the spectrum of disabilities they addressed in their compliance with the Web Content Accessibility Guidelines [4] (WCAG) and their respective levels of conformity.

Several recent studies have explored the impact of web accessibility on sustainability, highlighting the multifaceted dimensions of this relationship. A key aspect relates to social sustainability, where accessible web platforms promote equity and inclusion in access to critical health-related information and services [19]. This alignment with social sustainability principles emphasizes the importance of addressing the needs of diverse user groups, including those with disabilities, thereby fostering more equitable access to healthcare resources.

Furthermore, emerging trends highlight the environmental sustainability implications of accessible web platforms. As the digital ecosystem expands, digital services' energy consumption and carbon footprint have become pressing concerns. Research is increasingly delving into developing energy-efficient and environmentally friendly web accessibility

solutions that minimize the environmental impact of online health platforms [20]. These innovations improve the sustainability of digital health services and align with broader environmental sustainability goals.

Recent research also examines the economic sustainability aspect of web accessibility. Accessible health platforms can generate economic efficiencies by retroactively reducing the cost of meeting accessibility needs. This profitability contributes to the long-term sustainability of health services, ensuring their continued availability for all [21].

Additionally, with the increasing adoption of telemedicine and e-health services, the role of web accessibility in promoting healthcare sustainability is more pronounced than ever. These services, often delivered through digital platforms, must be universally accessible to all users, especially those in remote or underserved areas. The interplay between web accessibility, telehealth, and healthcare sustainability presents a critical area for further exploration.

## 3. Methods

The methodology used in this scoping review, which explores the accessibility of health platforms, has relevance in the broader field of sustainability. The accessibility assessment of health platforms not only encompasses a comprehensive review of compliance with international accessibility guidelines and assessments using manual and automated methods but also highlights the fundamental role of sustainability in this context. Sustainability in healthcare is intrinsically linked to equitable access and the distribution of resources and services.

By examining the sustainability aspects of web accessibility in the context of health platforms, this review indirectly addresses vital principles of sustainability.

This methodology sheds light on how accessible health platforms can promote social sustainability by ensuring inclusivity, economic sustainability through profitability, and environmental sustainability by driving energy-efficient solutions and reducing their digital carbon footprint.

This comprehensive approach aligns the study with emerging sustainability trends, fostering a deeper understanding of the dynamic relationship between web accessibility and sustainability in healthcare.

This research was devised as a scoping review (SR) designed to understand the accessibility landscape in health platforms comprehensively. The focus was on capturing a broad range of articles, highlighting the methodologies, tools, and guidelines for evaluating accessibility. For this, we employed the checklist of the PRISMA extension for scoping reviews (PRISMA-ScR) [22], adapting the method to detect accessibility-related papers regarding healthcare websites. Appendix B displays the checklist showing the number of pages following the compliance aspects suggested by the PRISMA-ScR (Title, abstract, introduction, methods, results, discussion, and funding).

The review protocol includes four steps: (1) the definition of the study approach and research questions, (2) the search strategy and quality evaluation, (3) the screening of studies, and (4) data extraction review and quality outcomes.

### 3.1. Study Approach and Research Questions

Investigating accessible websites for health is crucial as it helps ensure that reliable, relevant, and up-to-date health information is readily available to the public and vulnerable groups. The research question this study attempts to address is as follows: what factors contribute to the accessibility of health platforms? To answer this question, this study explores the existing literature related to the accessibility of healthcare platforms. This study identified the following factors that contribute to the accessibility of healthcare platforms: web accessibility standards that guide how to design and develop accessible healthcare platforms, accessibility tools and technologies, and accessibility awareness to help developers and healthcare providers create accessible healthcare platforms.

The primary objectives of this study are (1) to illustrate the information related to the most relevant research on accessibility in healthcare websites, including papers from different databases, authors, years of publication, and the SCImago Journal Rank (SJR) Impact factors; (2) identify the methods applied for the accessibility evaluation of healthcare websites over the years; and (3) monitor trends in the application of the WCAG in order to identify the existing barriers that may limit access to health information, especially for underserved populations.

Our study analyzes the results of relevant publications on accessibility in healthcare websites to address disparities in health information access and promote continuous improvement regarding accessibility. The research questions and motivation are presented in Table 1; they were defined to fulfill the aim of this SR. In order to determine the review scope, we apply the PCC method described by [23] to specify the key elements of the SR, facilitating the identification of essential research and establishing the inclusion and exclusion criteria of the study. PCC: Population (P), web accessibility; Concept (C), accessibility standards, guidelines, and parameters for health-related platforms; Context (C), healthcare-related digital platforms, healthcare-related environments.

**Table 1.** Research questions.

| No. | Research Question | Motivation |
|---|---|---|
| RQ1 | Which journals publish papers on the web accessibility of healthcare? | To evaluate the journals of the published papers. |
| RQ2 | What is the journal ranking for the included papers? | To examine the relevance and quality of the selected papers. |
| RQ3 | What is the frequency of publication of web accessibility studies in healthcare over time? | To understand the development of the publications over time. |
| RQ4 | What are the standards and disability guidelines used in the included papers? | To establish the standards and disability guidelines used in the included papers. |
| RQ5 | What empirical methods are used to evaluate the accessibility of health-related platforms? | To analyze the types of empirical validation methods used for this evaluation. |
| RQ6 | What type of online tools or services, real users, and experts have helped to evaluate website accessibility? | To recognize the accessibility assessment tools used in accessibility assessment. |
| RQ7 | What are the disabilities analyzed in the accessibility evaluations of healthcare platforms? | To determine the disabilities analyzed in accessibility studies about health websites. |
| RQ8 | What are the WCAG and conformance levels that have been used in the evaluation of healthcare platforms? | To identify the WCAG used in the evaluation of health platforms. |
| RQ9 | Does the document describe the errors or faults detected when evaluating the accessibility of a website? | To establish whether the article identifies the errors related to accessibility. |
| RQ10 | What are the results obtained from evaluating the accessibility of health-related platforms? | To understand the results obtained from the accessibility evaluation. |

### 3.2. Search Strategy and Quality Evaluation

The search string shown in Table 2 was designed to be of a broad scope and accessible size. The key terms used are linked to the research questions and have been selected based on the aim of this review and the PCC framework described above.

On 23 May 2023/20 September 2023, a systematic search was performed on several prominent databases, including the ACM Digital Library, IEEE Xplore, PubMed, ScienceDirect, Scopus, and the Web of Science (WOS). We adopted a multi-pronged search strategy, incorporating a combination of keywords such as "health platforms", "accessibility", "web accessibility", "evaluation tools", "WCAG", and "usability". The search parameters were designed to be inclusive to ensure that we captured a comprehensive range of relevant research.

**Table 2.** Search string.

| Database | Search String | Number of Studies |
|---|---|---|
| ACM Digital Library | [[All: web] OR [All: platforms]] AND [All: wcag] AND [All: health] | 20 |
| IEEE Xplore | (web OR platforms) AND wcag AND health | 8 |
| PubMed | (web OR platforms) AND wcag AND health | 15 |
| ScienceDirect | (web OR platforms) AND wcag AND health | 25 |
| Scopus | (web OR platforms) AND wcag AND health | 74 |
| Web of Science | (web OR platforms) AND wcag AND health | 21 |
| | Total | 163 |

The articles included in this review met the following criteria: They pertained directly to the accessibility of health platforms. They described their methodologies, either manual or automated, for evaluating accessibility. They mentioned using specific evaluation tools or adhering to accessibility guidelines like the WCAG [4]. They were written in English or Spanish and published in peer-reviewed journals.

The exclusion criteria comprised articles not directly related to health platforms or those related to accessibility only, articles that did not specify their methodology or tools for evaluation, non-English or -Spanish articles, and non-peer-reviewed literature.

To assess the significance of the selected papers and provide guidance for the interpretation of the findings and discussion of the results, we conducted a quality assessment (QA) [24] with five levels. A score of one means that the article complies, or zero if it does not comply, with the following criteria. (1) The paper relates to accessibility in health platforms; (2) the WCAG standards applied to accessibility evaluations in health platforms are described; (3) the article discusses the findings of accessibility evaluations; (4) limitations in the evaluated health platforms are described; (5) the journal or conference is indexed in the SJR. For the evaluation of quartiles, the SJR ranking website was used. The quality assessment scheme is shown in Table 3.

**Table 3.** Quality Assessment Scheme.

| No. | Quality Assessment Questions | Answer |
|---|---|---|
| QA1 | Does the paper relate to accessibility in health platforms? | (+1) Yes/(+0) No |
| QA2 | Does the paper specify the WCAG standards applied to accessibility evaluations of health platforms? | (+1) Yes/(+0) No |
| QA3 | Does the paper discuss any findings on the accessibility evaluations of health platforms? | (+1) Yes/(+0) No |
| QA4 | Are limitations described in the health platforms considered for accessibility assessments? | (+1) Yes/(+0) No |
| QA5 | Is the journal or conference publisher indexed in the SJR? | (+1) if it is ranked Q1, (+0.75) if it is ranked Q2, (+0.50) if it is ranked Q3, (+0.25) if it is ranked Q4, (+0.0) if it is not ranked. |

### 3.3. Screening of Studies

Following the inclusion and exclusion criteria defined above, the primary research included publications from between 2005 and 2023.

The Web Content Accessibility Guidelines (WCAG) [25] have played a crucial role in website accessibility since 1999, when the first version, the WCAG 1.0, was published [26]. A significant milestone occurred in 2008 with the release of the WCAG 2.0 [27], based on ISO/IEC 40500:2012 [28], an international standard in information technology. This



standard provides comprehensive guidelines to enhance web content accessibility for individuals with disabilities.

In 2018, the WCAG 2.1 [4] were introduced as the next step in evolving web accessibility standards. More recently, in 2023, the WCAG 2.2 [29] were released, representing the latest version of these guidelines. It is important to note that versions 2.0 and 2.1 remain recognized as essential web accessibility standards.

As a result, studies conducted before 2005 are not excluded from consideration, as they are still relevant for understanding the historical context and evolution of web accessibility guidelines and standards.

We applied the PRISMA-ScR tool [22] as it improves the quality and transparency in systematic reviews, avoiding bias. The screening and selection process is displayed in the flow diagram shown in Figure 1. This procedure includes the accessed databases, number of papers found per database, number of duplicates detected, and rejected papers according the inclusion and exclusion criteria. Finally, the number of included papers extracted for evaluation is presented.

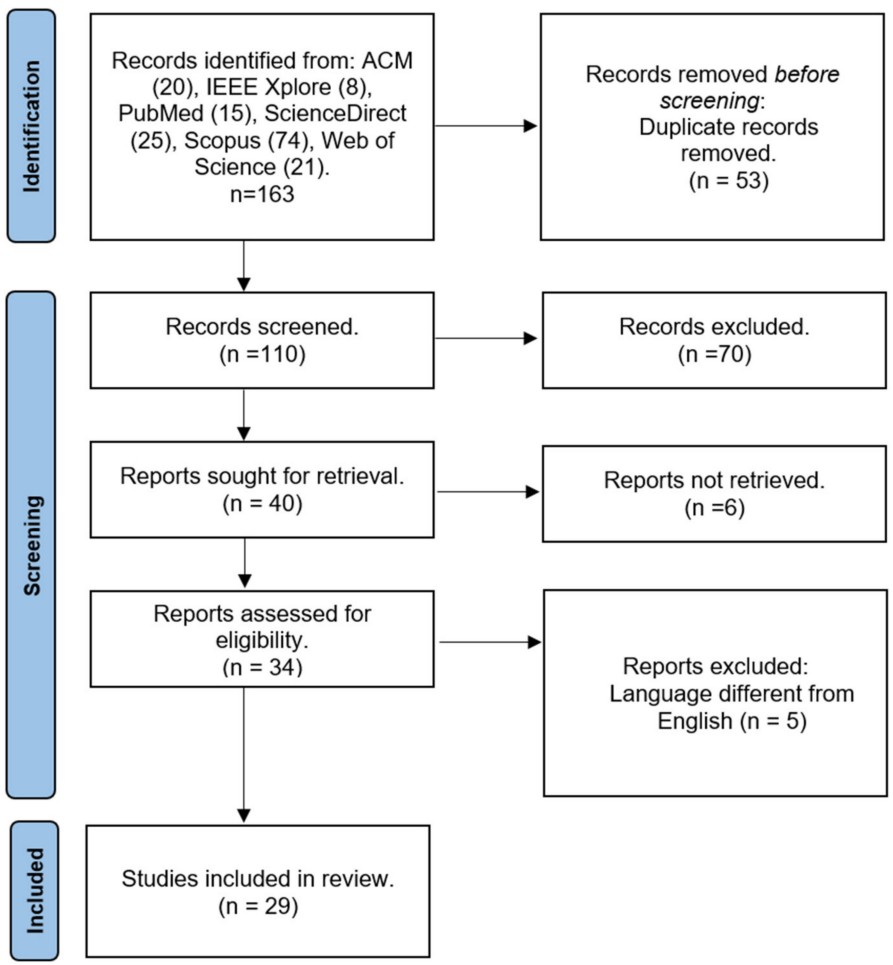

**Figure 1.** PRISMA flow diagram for the scoping review.

Phase 1: Identification. The registers obtained from the database searches were 20 documents from the ACM Digital Library, eight from IEEE Xplore, 15 from PubMed, 25 from ScienceDirect, 74 from Scopus, and 21 from WOS; 163 papers were extracted.

Phase 2: Screening. Following the inclusion and exclusion criteria, from 146 articles, 53 were removed as duplicates, and 110 articles continued to be screened. From these, 70 papers irrelevant to accessibility in health platforms were excluded, and 40 studies were incorporated. Then, six reports had unavailable full texts and were thus removed.

Next, five were written in another language and excluded out of the 34 reports assessed for eligibility.

Phase 3: Included. The 29 remaining articles were approved for inclusion in the review process. The authors performed a full-text review of the included articles focused on primary accessibility studies in healthcare platforms. We assessed the publication's quality according to the criteria detailed in Table 3. Cohen's kappa coefficient was applied to indicate the degree of reliability in classifying the included articles. A value of 0.613 was obtained with a percentage of 80.8%, resulting in a substantial agreement between the evaluators.

### 3.4. Data Extraction Review and Quality Outcomes

From the selected articles, relevant data were meticulously extracted and categorized. These data included the tools used for evaluation, whether automated like Achecker, WAVE, or TAW, or manual methods such as user feedback, expert consultations, and task-based validations; the range of disabilities addressed in the evaluations; adherence to the Web Content Accessibility Guidelines (WCAG); and the mentioned conformance levels.

After we obtained the pertinent data from the ACM Digital Library, IEEE Xplore, ScienceDirect, Scopus, and WOS databases, the information was transferred into the StArt-Lapes tool, version 3.3, to process the selected papers by eliminating duplicates, screening the reports assessed for eligibility, and including the selected papers for review.

Once the data were extracted, they were synthesized and analyzed to identify patterns, trends, and critical insights. We specifically looked at the distribution of articles using automated and manual methods versus those employing only one type; the frequency of use of various evaluation tools and the progression and adoption of versions of the WCAG and their conformance levels over time.

While scoping reviews aim to provide a broad overview rather than a deep evaluation, an essential quality assessment (QA) was performed to ensure the relevance and credibility of the included articles using the five questions presented in Table 3. The QA outcomes of the quality assessment are detailed in Table 4, including the paper ID, publication's title, score applied to the five criteria, and normalization values from 0 to 1. The normalization values were calculated by applying the following equation:

$$Normalization = \frac{Score - \min(Score)}{[\max(Score) - \min(Score)]} \tag{1}$$

Given that the nature of this study was a scoping review, there was no direct involvement of human participants. However, all the articles and their findings were treated with the utmost respect and confidentiality.

**Table 4.** Scientific articles included and quality assessment results.

| ID | Publication Name | Quartile | SJR Factor | QA1 | QA2 | QA3 | QA4 | QA5 | Score | Normalization |
|---|---|---|---|---|---|---|---|---|---|---|
| JiA22 | Investigation of COVID-19 Vaccine Information Websites across Europe and Asia Using Automated Accessibility Protocols | Q2 | 0.83 | 1 | 1 | 1 | 1 | 0.75 | 4.75 | 0.8 |
| TeN22 | Accessibility of COVID-19 Websites of Asian Countries: An Evaluation Using Automated Tools | Q2 | 0.6 | 1 | 1 | 1 | 1 | 0.75 | 4.75 | 0.8 |
| NaA22 | Evaluating the accessibility of public health websites: An exploratory cross-country study | Q2 | 0.75 | 1 | 1 | 1 | 1 | 0.75 | 4.75 | 0.8 |
| NaB22 | Implementation of e-learning Platform for Increasing Digital Health Literacy as a Condition for Integration of e-health Services with PHR | N/A | 0 | 1 | 1 | 1 | 1 | 0.00 | 4.00 | 0.0 |



**Table 4.** *Cont.*

| ID | Publication Name | Quartile | SJR Factor | QA1 | QA2 | QA3 | QA4 | QA5 | Score | Normalization |
|---|---|---|---|---|---|---|---|---|---|---|
| AnB22 | Implementation of Innovative e-Health Services and Digital Healthcare Ecosystem Cross4all Project Summary | N/A | 0.2 | 1 | 1 | 1 | 1 | 0.00 | 4.00 | 0.0 |
| SeM22 | Accessibility evaluation of university hospital websites in Turkey | Q2 | 0.75 | 1 | 1 | 1 | 1 | 0.75 | 4.75 | 0.8 |
| KuS22a | Accessibility and Performance Evaluation of Healthcare and E-Learning Sites in India: A Comparative Study Using TAW and GTMetrix | Q4 | 0.17 | 1 | 1 | 1 | 1 | 0.25 | 4.25 | 0.3 |
| KuS22b | Accessibility of Healthcare Sites: Evaluation by Automated Tools | Q4 | 0.15 | 1 | 1 | 1 | 1 | 0.25 | 4.25 | 0.3 |
| GlA21 | Improvement of Accessibility in Medical and Healthcare Websites | Q4 | 0.15 | 1 | 1 | 1 | 1 | 0.25 | 4.25 | 0.3 |
| PaA21 | Challenges of Web Accessibility in a Health Application to Predict Neonatal Mortality—The Score Bebe ® | Q4 | 0.15 | 1 | 1 | 1 | 1 | 0.25 | 4.25 | 0.3 |
| ElF20 | We are exploring who communication during the COVID-19 pandemic through the Who website based on W3C guidelines: Accessible for all? | Q2 | 0.83 | 1 | 1 | 1 | 1 | 0.75 | 4.75 | 0.8 |
| MuA20 | Effectiveness of web accessibility policy implementation in online healthcare information | Q3 | 0.29 | 1 | 1 | 1 | 1 | 0.50 | 4.5 | 0.5 |
| PaA20a | Web Accessibility Analysis of a Tele-Rehabilitation Platform: The Physiotherapist Perspective | Q4 | 0 | 1 | 1 | 1 | 1 | 0.25 | 4.25 | 0.3 |
| PaA20b | Designing an Accessible Website for Palliative Care Services | Q4 | 0.19 | 1 | 1 | 1 | 1 | 0.25 | 4.25 | 0.3 |
| PaA20c | Challenges and improvements in website accessibility for health services | Q4 | 0 | 1 | 1 | 1 | 1 | 0.25 | 4.25 | 0.3 |
| PaA20d | Improving web accessibility: Evaluation and analysis of a telerehabilitation platform for hip arthroplasty patients | Q4 | 0 | 1 | 1 | 1 | 1 | 0.25 | 4.25 | 0.3 |
| CeS19 | Accessibility Testing of European Health-Related Websites | Q1 | 0.48 | 1 | 1 | 1 | 1 | 1.00 | 5 | 1.0 |
| LuC19 | Web accessibility of Internet appointment scheduling in primary care | Q3 | 0.47 | 1 | 1 | 1 | 1 | 0.50 | 4.5 | 0.5 |
| PaA18a | Towards Web Accessibility in Telerehabilitation Platforms | N/A | 0 | 1 | 1 | 1 | 1 | 0.00 | 4 | 0.0 |
| PaA18b | Framework for Accessibility Evaluation of Hospital Websites | N/A | 0 | 1 | 1 | 1 | 1 | 0.00 | 4 | 0.0 |
| JoM17 | A full-scope web accessibility evaluation procedure proposal based on Iberian eHealth accessibility compliance | Q1 | 2.46 | 1 | 1 | 1 | 1 | 1.00 | 5 | 1.0 |
| ArK17 | Evaluating the accessibility, usability, and security of Hospitals websites: An exploratory study | N/A | 0 | 1 | 1 | 1 | 1 | 0.00 | 4 | 0.0 |
| EdL15 | My health: An online healthcare social network inclusive for elderly people | N/A | 0 | 1 | 1 | 1 | 1 | 0.00 | 4 | 0.0 |
| LaO05 | Accessibility compliance rates of consumer-oriented Canadian health care Web sites | Q1 | 0.74 | 1 | 1 | 1 | 1 | 1.00 | 5 | 1.0 |
| GrB22 | COVID-19 vaccine website accessibility dashboard | Q1 | 1.64 | 1 | 1 | 1 | 1 | 1.00 | 5 | 1.0 |
| SaA21 | Accessibility evaluation of COVID-19 vaccine registration websites across the United States | Q1 | 2.44 | 1 | 1 | 1 | 1 | 1.00 | 5 | 1.0 |
| NoY18 | Website Accessibility: U.S. Veterans Affairs Medical Centers as a Case Study | Q2 | 0.48 | 1 | 1 | 1 | 1 | 0.75 | 4.75 | 0.8 |
| NoY21 | The accessibility of state occupational safety and health consultation websites | Q2 | 0.75 | 1 | 1 | 1 | 1 | 0.75 | 4.75 | 0.8 |

## 4. Results

This section answers the ten research questions proposed in Table 1, first via a bibliometric analysis comprising relevant information such as the type of journal and publications

over time. Then, the SR is described to map the articles in line with the accessibility of health platforms.

### 4.1. Bibliometric Analysis

### 4.1.1. RQ1: Which Journals Publish Papers on the Web Accessibility of Healthcare?

The data extracted indicate that 15 journals publish papers on the web accessibility of healthcare. Additionally, other information on this matter is found in conference articles published as part of book series, in workshops, and proceedings. Figure 2 shows that the journal UAIS has the most significant number of publications with four papers published, followed by the IJERP with two papers. Additionally, among the book series and other conference publishers, LNNS and AISC appear to have the most publications with three papers, respectively, while ACMSIG has two papers published.

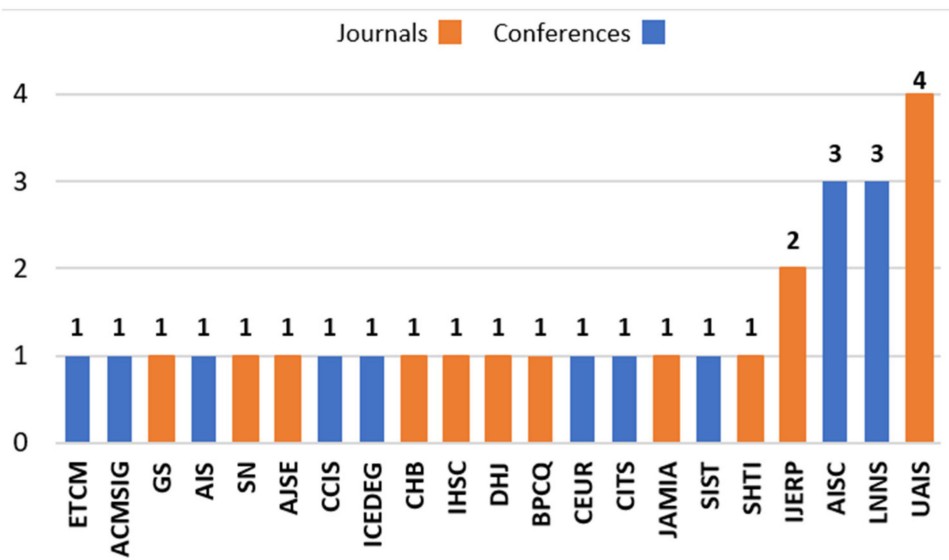

**Figure 2.** RQ1-extracted data from journals with publications on web accessibility of healthcare.

The remaining six journals and conferences have only one included paper each (see detailed information in Table A1).

The analyzed data also include the countries from which the journals or conferences are derived, Figure 3. Through the SJR website, we found that, from the 21 journals and conferences in which the selected papers were published, 11 papers were from Germany and the United States, respectively, followed by Switzerland with five papers, and the United Kingdom with two papers; Spain and the Netherlands each had one publication.

Furthermore, regarding the countries where the healthcare websites were studied, Figure 3 reveals the United States as the leading country, with four papers on this topic, followed by Hungary with two papers; Canada, Finland, Jordan, Portugal, South Korea, and Turkey each have one publication, respectively, in journals. On the other hand, the leading country in publishing conference papers is Ecuador, with eight conferences; then Macedonia, with two; and then Brazil, with one publication. Finally, the only country with both articles and conferences appear to be India, with one article and three conferences, while Spain has two conferences.

### 4.1.2. RQ2: What Is the Journal Ranking for the Included Papers?

The inclusion of sustainability in the context of the scientific journal rankings of the selected articles highlights the relevance of addressing accessibility in health from a sustainable perspective.

The ranking of its publishing journal can measure a paper's acceptability. From the 29 selected papers, 23 articles were published in SJR-ranked journals. As shown in Figure 4,

the largest percentage of papers was equally distributed among Q2 and Q4, with 28% articles each. Unranked publishers published 21%, only 17% of papers were ranked in Q1, and 7% of the selected papers were Q3. Quartiles from the SJR journals were obtained following the year of publication of each paper. As the final year of publication considered in this study is 2023, all the quartiles consulted are reported.

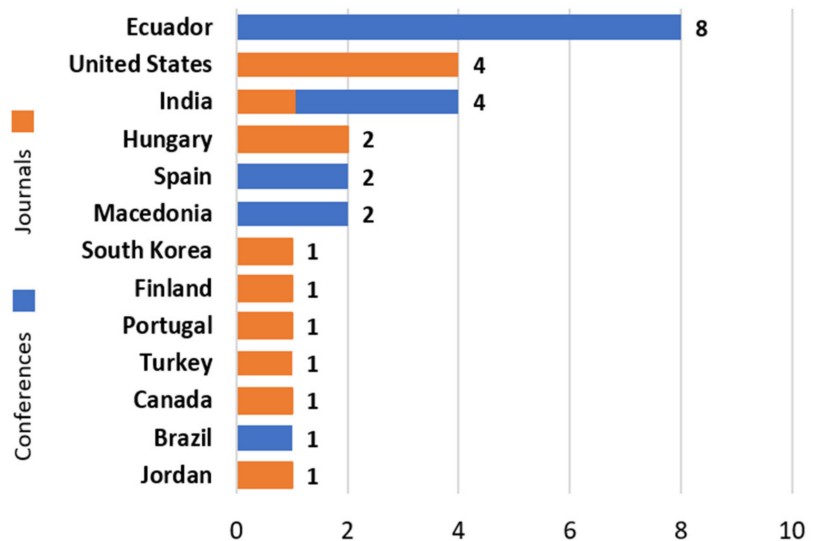

**Figure 3.** RQ1 extracted data, including the number of papers per country.

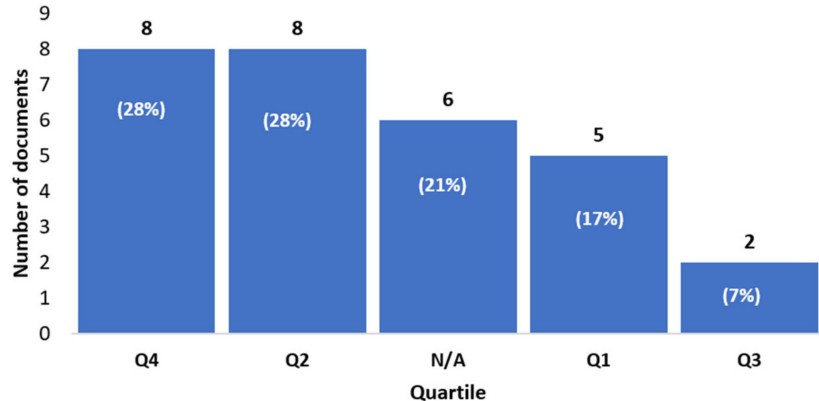

**Figure 4.** Number of documents per quartile indexed in SJR.

4.1.3. RQ3: What Is the Frequency of Publication of Web Accessibility Studies in Healthcare over Time?

The importance of sustainability in the frequency of web accessibility studies in healthcare over time is evident in the analysis of the articles published during the years included in this review. It was observed that the largest number of publications was found in the most recent years, with a significant peak in 2022, with nine articles. This frequency could reflect a growing awareness of the importance of addressing accessibility in healthcare platforms from a sustainable perspective.

The publication years of the included papers range from 2005 to 2022, as displayed in Figure 5. As observed, the highest number of publications are found in the year 2022 with nine papers, and then 2021 with seven papers, 2021 with four, 2018 with three, 2017 and 2019 with two papers each, and 2005 and 2015 with one publication each (detailed information in Table A1 and Appendix A).

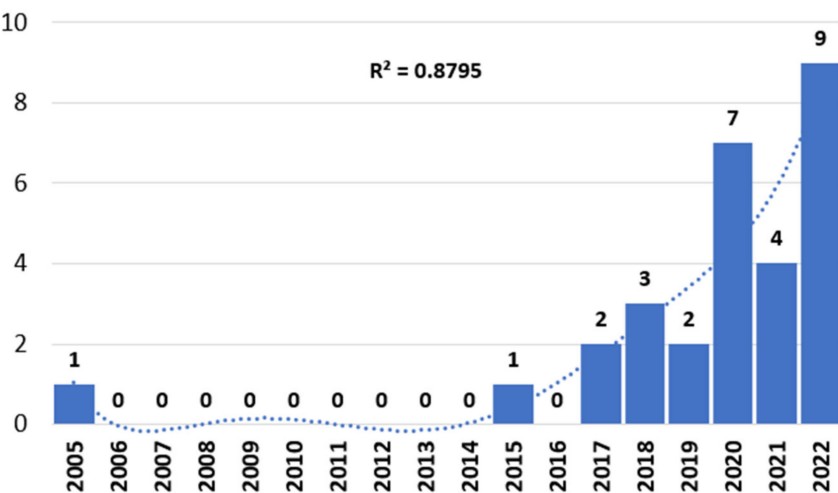

**Figure 5.** Number of publications per year.

### 4.2. Scoping Review

Table 5 presents the 29 included papers organized from the most to the least recent year of publication. The list includes the assigned paper ID (generated using the first word of the first author's surname and year of publication), the article's title, the first author, and the reference, followed by the year of publication.

**Table 5.** List of scientific publications included in this review.

| No. | Paper ID | Title | Reference | Year |
| --- | --- | --- | --- | --- |
| 1 | JiA22 | Investigation of COVID-19 Vaccine Information Websites across Europe and Asia using Automated Accessibility | Ara, J. [30] | 2022 |
| 2 | TeN22 | Accessibility of COVID-19 Websites of Asian Countries: An Evaluation Using Automated Tools | Niom, T. [31] | 2022 |
| 3 | NaA22 | Evaluating the accessibility of public health websites: An exploratory cross-country study | Alajarmeh, N. [32] | 2022 |
| 4 | NaB22 | Implementation of e-learning Platform for Increasing Digital Health Literacy as a Condition for Integration of e-health Services with PHR | Blazheska-Tabakovska, N. [33] | 2022 |
| 5 | AnB22 | Implementation of Innovative e-Health Services and Digital Healthcare Ecosystem Cross4all Project Summary | Bocevska, A. [34] | 2022 |
| 6 | SeM22 | Accessibility evaluation of university hospital websites in Turkey | Macakoğlu, Ş. [35] | 2022 |
| 7 | KuS22a | Accessibility and Performance Evaluation of Healthcare and E-Learning Sites in India: A Comparative Study Using TAW and GTMetrix | Sarita, K. [36] | 2022 |
| 8 | KuS22b | Accessibility of Healthcare Sites: Evaluation by Automated Tools | Sarita, K. [15] | 2022 |
| 9 | GrB22 | COVID-19 vaccine website accessibility dashboard | Jo, G. [37] | 2022 |
| 10 | GlA21 | Improvement of Accessibility in Medical and Healthcare Websites | Acosta-Vargas, G. [38] | 2021 |
| 11 | PaA21 | Challenges of Web Accessibility in a Health Application to Predict Neonatal Mortality: The Score Bebe Â® | Acosta-Vargas, P. [39] | 2021 |
| 12 | SaA21 | Accessibility evaluation of COVID-19 vaccine registration websites across the United States | Alismail, S. [40] | 2021 |
| 13 | NoY21 | The accessibility of state occupational safety and health consultation websites | Youngblood, N. [41] | 2021 |

**Table 5.** *Cont.*

| No. | Paper ID | Title | Reference | Year |
|---|---|---|---|---|
| 14 | ElF20 | I am exploring WHO communication during the COVID-19 pandemic through the who website based on W3C guidelines: Accessible for all? | Fernández-Díaz, E. [42] | 2020 |
| 15 | MuA20 | Effectiveness of web accessibility policy implementation in online healthcare information | Arief, M. [43] | 2020 |
| 16 | YoJ20 | Web accessibility of healthcare Web sites of Korean government and public agencies: a user test for persons with visual impairment | Yi, Y.J. [44] | 2020 |
| 17 | PaA20a | Web Accessibility Analysis of a Tele-Rehabilitation Platform: The Physiotherapist Perspective | Acosta-Vargas, P. [45] | 2020 |
| 18 | PaA20b | Designing an Accessible Website for Palliative Care Services | Acosta-Vargas, P. [46] | 2020 |
| 19 | PaA20c | Challenges and improvements in website accessibility for health services | Acosta-Vargas, P. [47] | 2020 |
| 20 | PaA20d | Improving web accessibility: Evaluation and analysis of a telerehabilitation platform for hip arthroplasty patients | Acosta-Vargas, P. [48] | 2020 |
| 21 | CeS19 | Accessibility Testing of European Health-Related Websites | Sik-Lanyi, C. [49] | 2019 |
| 22 | LuC19 | Web accessibility of Internet appointment scheduling in primary care | Casasola Balsells, L.A. [50] | 2019 |
| 23 | PaA18a | Towards Web Accessibility in Telerehabilitation Platforms | Acosta-Vargas, P. [51] | 2018 |
| 24 | PaA18b | Framework for Accessibility Evaluation of Hospital Websites | Acosta-Vargas, P. [52] | 2018 |
| 25 | NoY18 | Website Accessibility: U.S. Veterans Affairs Medical Centers as a Case Study | Youngblood, N. [53] | 2018 |
| 26 | JoM17 | A full-scope web accessibility evaluation procedure proposal based on Iberian eHealth accessibility compliance | Martins, J. [54] | 2017 |
| 27 | ArK17 | Evaluating the accessibility, usability, and security of Hospitals websites: An exploratory study | Kaur, A. [55] | 2017 |
| 28 | EdL15 | My health: An online healthcare social network inclusive for elderly people | Medina, E.L. [56] | 2015 |
| 29 | LaO05 | Accessibility compliance rates of consumer-oriented Canadian healthcare Web sites | O'Grady, L. [57] | 2005 |

4.2.1. RQ4: What Are the Standards and Disability Guidelines Used in the Included Papers?

Of the 29 selected articles, regardless of the version, all papers used the WCAG for their accessibility evaluations of healthcare platforms, as shown in Figure 6. Although the WCAG is a globally recognized set of guidelines to help make web content more accessible to disabled people [58], this study shows that it is not always the only standard used to evaluate the accessibility of health-related websites. It is to be noted that three papers use Section 508, a United States' federal law, to ensure access to digital information for disabled people [59]. Additionally, one paper uses ISO 9241-171: 2008, a set of standards that provide ergonomic specifications and guidance for designing accessible software used at home, work, education, and in public places, intended to help people with disabilities ranging from physical, sensory, cognitive, and motor disabilities to elderly and temporarily disabled users [60].

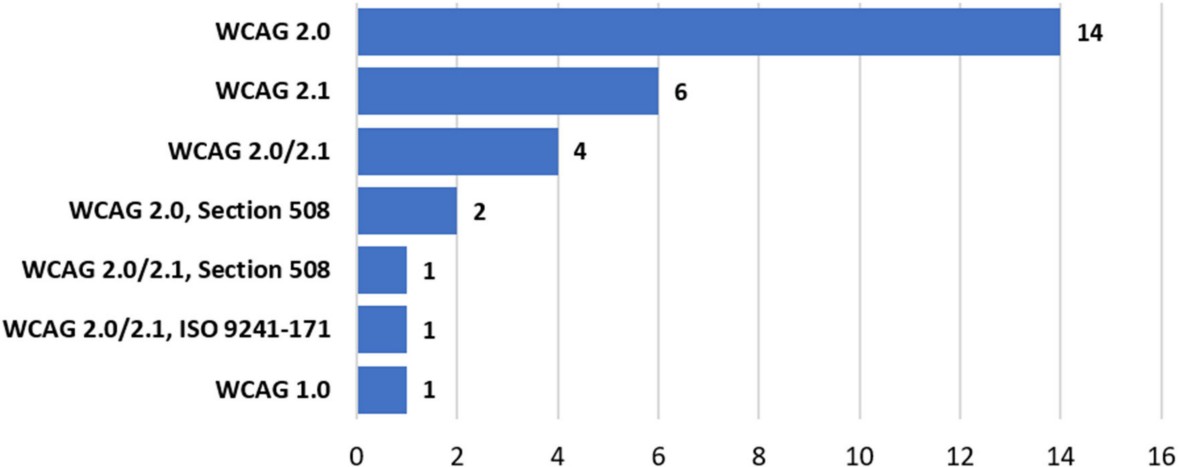

**Figure 6.** Number of papers per accessibility standard described.

Sustainability in this context means that the standards and guidelines must be maintained and evolve to ensure that digital healthcare remains accessible and usable for all, promoting equity and inclusion.

The choice of the accessibility standards and guidelines used in digital healthcare platforms directly impacts the sustainability of web accessibility in the healthcare sector. The evolution and maintenance of these standards are crucial to ensure that people with disabilities continue to have equitable access to online health information and services.

4.2.2. RQ5: What Empirical Methods Are Used to Evaluate the Accessibility of Health-Related Platforms?

Of the methods described in the publications for the accessibility evaluation of healthcare websites, three standard methods have been described: (1) Automatic methods, used by 52% of the articles. (2) Manual methods, including qualitative, quantitative, statistical, expert, and real-user validation, in 14% of papers. (3) Automatic and manual methods: 34% of the papers combine both. Sustainability in this context also relates to the need to continuously adapt and improve assessment methods to address changing challenges in the digital healthcare environment. Figure 7 shows the papers according to the method of evaluation described (for complete information see Table A2—Appendix A.

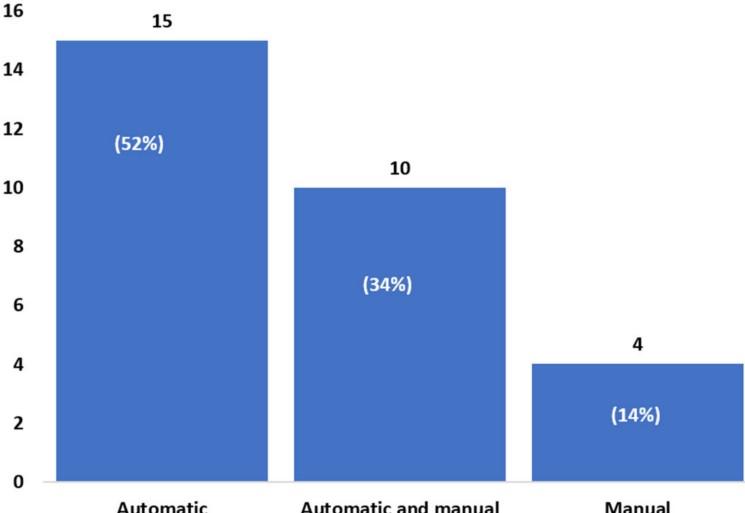

**Figure 7.** Number of papers per empirical method used.

The choice of methods, whether automatic or manual, and the continuous evolution of these methods are fundamental to maintaining the sustainability of web accessibility in the healthcare sector. Sustainability in this context is closely related to the choice and evolution of the empirical methods used to evaluate the accessibility of healthcare platforms. This question helps to ensure that people with disabilities continue to access online healthcare effectively and equitably as technologies and healthcare practices evolve.

### 4.2.3. RQ6: What Type of Online Tools or Services, Real Users, and Experts Have Helped to Evaluate Website Accessibility?

In order to obtain a robust website accessibility evaluation, an automatic assessment tool needs to be applied hand in hand with the manual review of an expert. As these automated or online tools can generate mistaken or inaccurate results, real-user and expert validation is critical [61]; to this end, Figure 8 shows that ten of the 29 included articles used real users' and expert validation. Of this group, six articles used manual evaluation and automatic assessment tools, with the last four using only a manual review. The remaining 19 articles apply automatic evaluation tools without specifying whether a manual review was conducted (user or expert).

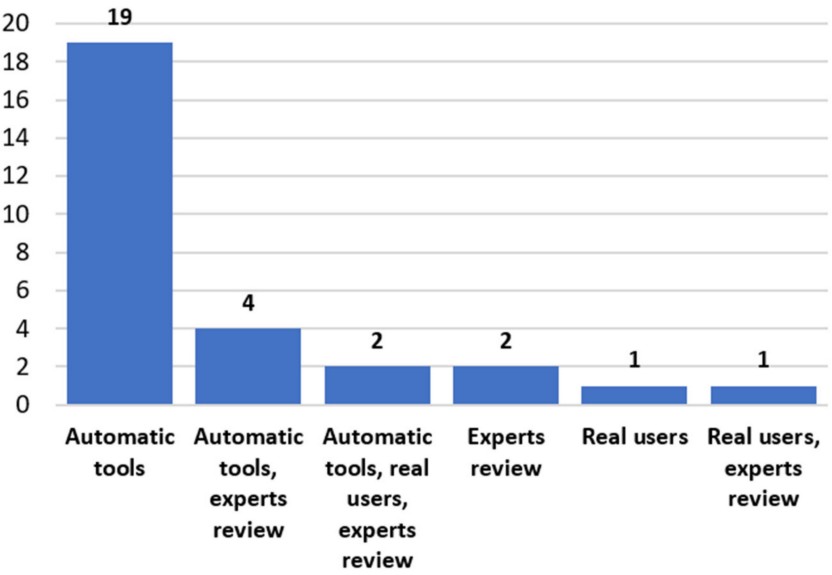

**Figure 8.** The number of publications by the method applied.

In the six articles using both types of evaluation, we found that manual review tools like user task-based validation, expert web consultors, user feedback, questionnaires, expert review for readability score, and language analysis were used; on the other side, automated review tools like Mauve++, Nibbler, WAVE, Achecker, SortSite, TAW, and others were applied. In the articles that used only manual evaluation tools, we found that the papers described expert reviews, user evaluation with task-based tests on the overall difficulty level to complete a task, interviews, accessibility checklists, usability tests, and questionnaires.

On the other hand, the automatic evaluation tools found within the articles analyzed were Access Monitor, Achecker, WAVE, CSS, TAW, Deadlink Checker, Google Mobile Test GTMetrix, Axe, X (HTML), eXaminator, Siteimprove, OpenWAX, Tenon, BobbyTM, SortSite, and WAO. Moreover, across the 29 included articles, there were three most commonly used automatic tools for accessibility evaluations; more of their functionalities are described below. Figure 9 synthesizes the manual and automatic tools used more than once in the selected articles for website accessibility evaluations (for complete information, see Table A2).

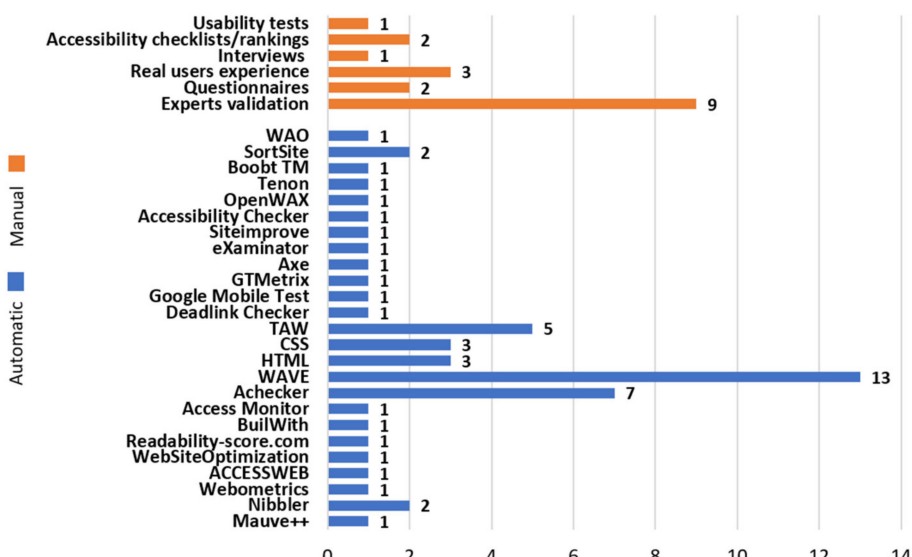

**Figure 9.** Documents by the method applied.

(1) Achecker: The accessibility checker is a digital tool for website evaluation based on the WCAG; it is used to rapidly scan the website of interest to identify the technical accessibility errors and issues it presents [31].

(2) WAVE: this comprises a set of evaluation instruments designed to enhance the accessibility of web content for individuals with disabilities; it not only identifies numerous accessibility and WCAG errors, but it is also centered on addressing issues known to affect end users, promoting human assessment, and educating about web accessibility [16].

(3) TAW: The web accessibility test (Test de Accesibilidad Web in Spanish) developed to verify the level of compliance with accessibility requirements of web pages against the WCAG 1.0, 2.0, and 2.1 [15]; the final report includes a list of problems, warnings, and non-reviewed items [62]. The Spanish Foundation Centre created it to develop Information and Communication Technologies in Asturias (CTIC) [63]. The sustainable approach implies that this combination of assessment methods is a constant practice and that the tools and services are up-to-date and adaptable. The analysis also highlights the most common automatic assessment tools in the included articles. To ensure sustainability in this context, these tools must remain effective as web technologies and accessibility guidelines evolve.

Sustainability in this context relates to ensuring that accessibility assessments are reliable and accurate over time. Using automated or online tools to assess the accessibility of healthcare platforms is a critical practice, but these tools can generate erroneous or inaccurate results in some cases. Therefore, validation by real users and experts is essential to ensure the accuracy of accessibility assessments.

### 4.2.4. RQ7: What Are the Disabilities Analyzed in the Accessibility Evaluations of Healthcare Platforms?

Each of the papers discusses disabilities; however, only 14 articles delineate the specific disabilities that they are focusing on. Some of the disabilities mentioned are visual disabilities like low vision, blindness, second-level sight-impaired people, auditory disabilities, physical disabilities, speech problems, cognitive/neurological problems, elderly users, and people with motor impairments. Additionally, we found six papers that emphasized analyzing web accessibility not just for disabled users by definition but also for anyone related to patients with specific conditions such as arthroplasty, chronic diseases, and palliative care patients. It is important to note that even if this type of user is not defined as a disabled user, they belong to a vulnerable group that should be considered. Other groups mentioned were users related to newborn health and children (for complete information see Table A2).

Thus, accessibility should not be limited to disabled users but should consider people with other conditions that might be unintentionally neglected.

The issue addressed in research question RQ7, related to the disabilities analyzed in the accessibility evaluations of healthcare platforms, also has a bearing on sustainability in the context of the web accessibility of healthcare.

Sustainability relates to ensuring that online health platforms are accessible and usable for a broad spectrum of users, including those with specific disabilities and health conditions. By considering and analyzing a variety of disabilities and health conditions in accessibility assessments, sustainability is promoted by ensuring that healthcare platforms are inclusive and able to meet the needs of all users.

Furthermore, the study mentions that accessibility should not be limited to disabled users but should consider people with other conditions who may be unintentionally marginalized. This topic is critical to promote sustainability by ensuring that no one is excluded from online healthcare because of their disability or specific health condition.

4.2.5. RQ8: What Are the WCAG and Conformance Levels That Have Been Used in the Evaluation of Healthcare Platforms?

As detailed in Figure 10, the papers evaluating accessibility in health-related websites have followed the corresponding updated versions of the WCAG through the years; the conformance levels are also described below (for complete information, see Table A3).

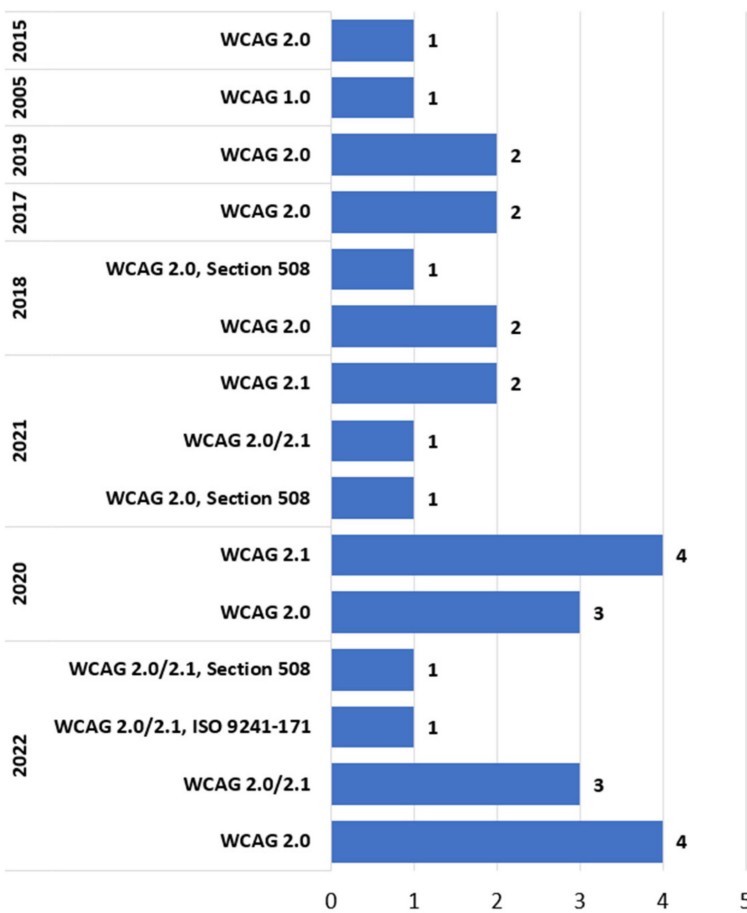

**Figure 10.** The WCAG applied by year of publication.

The first version, the WCAG 1.0, launched in 1999 [27], was applied by article [57], which did not specify the conformance level achieved. On the other hand, the WCAG 2.0 was launched in 2008; six publications applied this update starting from 2015 to 2018, before the last version was published. Paper [56] did not specify the level of conformance; paper [53] specified levels A and AA; papers [51,52,54,55] specified levels A, AA, and AAA.

As of June 2018, the latest published version was the WCAG 2.1; papers ranging from 2019 to 2022 include this update, along with the previous version of the WCAG, 2.0. It is noted that of both the papers published in 2019, one [50] applies the 2.0 version but also the first version 1.0, with conformance levels A, AA, and AAA, and the other [49] applies only the 2.0 update, although the latest version was already available by that year. Furthermore, between 2019 and 2022, we found that six papers applied a combination of WCAG versions 2.0 and 2.1; studies [30–34,40], and all evaluated conformance levels as A, AA, and AAA.

In analyzing the WCAG versions and conformance levels used in healthcare platform evaluations, sustainability is reflected in the importance of keeping up with the latest web accessibility guidelines. As web technologies evolve and user needs change, accessibility assessments must remain relevant and practical. This analysis ensures that healthcare platforms continue to meet accessibility standards over time and adapt to new guidelines as needed to maintain their sustainability in terms of accessibility.

The WCAG 2.0 update alone was still used in 16 documents in those years; the conformance levels evaluated were not specified in papers [37,43,44]; conformance level AA in paper [15]; and conformance levels A, AA, and AAA in papers [35,36,41,46,49].

Finally, only six documents applied the 2.1 version without any other combination; papers [45,47] did not specify the conformance level; article [39] specified level AA; study [38] specified level A and AA; and article [42,48] specified levels A, AA, and AAA.

Research question RQ8, related to WCAG versions and conformance levels used in evaluating healthcare platforms, relates to sustainability in the context of the web accessibility of healthcare.

Sustainability relates to the need to keep web accessibility guidelines updated and adapt accessibility assessments to the latest guidelines. Looking at how accessibility assessments have followed WCAG updates over time emphasizes the importance of keeping up with the latest standards and guidelines. This topic is essential to ensure that online health platforms remain accessible as web technologies evolve and user needs change.

In addition, this study shows how some accessibility assessments have applied multiple levels of conformance, including A, AA, and AAA levels. This variety reflects the importance of accommodating various needs and disabilities. This study promotes sustainability by ensuring that healthcare platforms are inclusive and accommodate diverse users.

### 4.2.6. RQ9: Does the Document Describe the Errors or Faults Detected When Evaluating the Accessibility of a Website?

The number of included articles describing errors according to the principles of the WCAG 2.0 and 2.1 is presented in Figure 11, with detailed information in Table A4. The most significant number of papers with errors found have errors related to the perceivable category of the WCAG, specifically in the 1.1.1 non-text content success criteria, with 39% of the articles mentioning errors (see the complete information in Table A4); this case is followed by operable and understandable categories, with 23% errors, and the robust category, with the lowest % error rate of 13%.

According to updates of the WCAG available over time, as shown in Figure 10, it is noted that for this research, just one document describes the application of the WCAG 1.0. Hence, errors of Priority 1 (Level A) are present once throughout the entire study, which means that the website analyzed in the included papers has followed the available updates from the W3C.

Regarding the WCAG 2.0 and 2.1, we found that the level of conformance with the most significant number of errors was level A, with 70% of errors occurring in the 29 papers assessed, level AA followed with 21%, and level AAA with 9% occurrence (for complete information see Table A4).

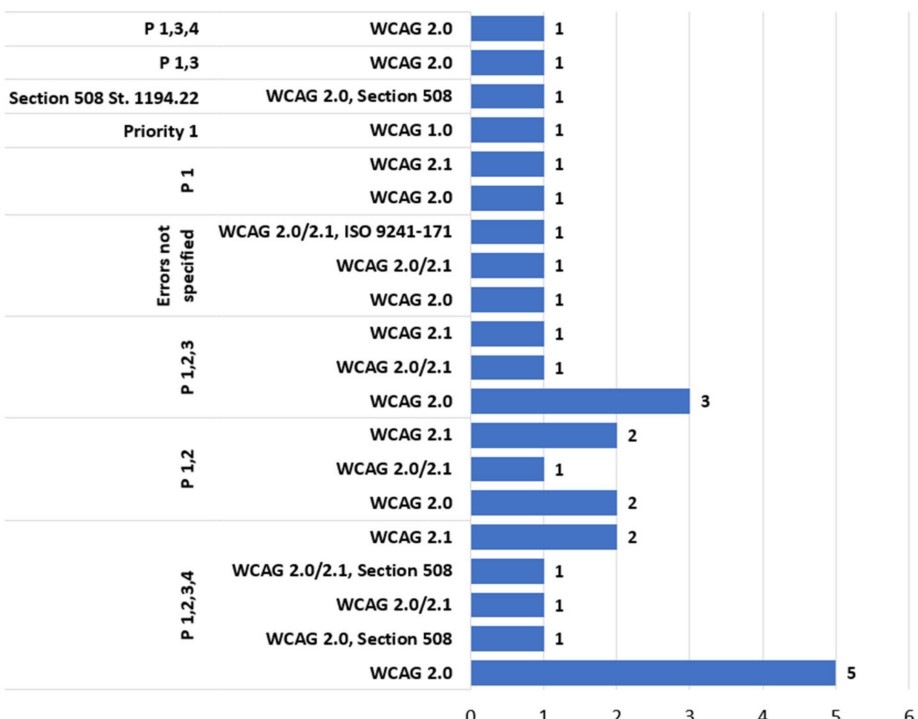

**Figure 11.** Number of papers per error described according to the WCAG, (1) perceivable, (2) operable, (3) understandable, and (4) robust, WCAG 1.0 (Priority 1), Section 508 Standard 1194.22 (A) [64].

In terms of sustainability, it is critical to address these accessibility errors and failures continuously to ensure that healthcare platforms remain accessible over time. This analysis involves making constant corrections and improvements to comply with the WCAG and ensure that platforms are sustainable. In addition, error analysis across the different WCAG versions highlights the importance of keeping updated with the latest web accessibility guidelines to address and prevent constantly evolving errors. This study contributes to the sustainability of healthcare platforms in terms of their ability to adapt to changes in accessibility guidelines and ensure equitable access to healthcare information and services.

Research question RQ9, related to the description of errors or bugs detected when assessing the accessibility of a website, relates to sustainability in the context of the need to continually address and correct accessibility errors to ensure that online health platforms remain accessible over time. Identifying and describing errors in accessibility assessments are essential to maintaining the sustainability of these platforms.

### 4.2.7. RQ10. What Are the Results Obtained from Evaluating the Accessibility of Health-Related Platforms?

The advantage of using the WCAG in its available and updated versions (2.0, 2.1, 3.0) is that this standard is used and applied internationally, making it easier to understand and adapt to current technological developments. As seen in Table A3, the 29 papers analyzed use the WCAG 1.0, 2.0, and 2.1 to evaluate the different health websites. Figure 12 shows that 38% of the analyzed websites correspond to healthcare websites in general (public or governmental). Followed by eHealth or e-learning websites being analyzed 21% of the time.

The final analyzed types were university hospital websites, the WHO website, alongside two different types of websites (e-learning and healthcare websites). Additionally, as presented in Table 6, regarding the number of websites analyzed per source type, hospital websites formed the most significant number, with 401 web pages assessed.

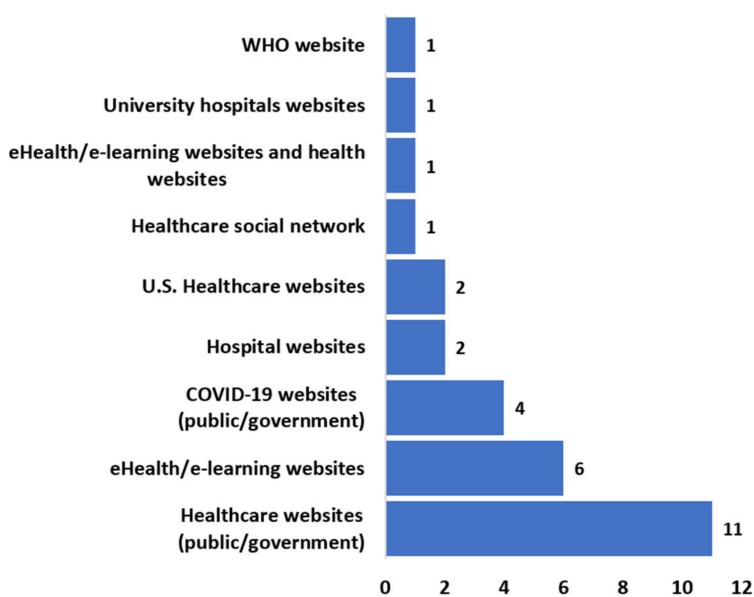

**Figure 12.** Number of articles by type of health websites.

**Table 6.** The number of websites analyzed per source.

| Sources | No. Analyzed Websites |
|---|---|
| Healthcare social networks | 1 |
| Hospital websites | 401 |
| U.S. Healthcare websites | 166 |
| University hospitals' websites | 58 |
| WHO website | 6 |
| COVID-19 websites (public/government) | 176 |
| Healthcare websites (public/government) | 270 |
| eHealth/e-learning websites | 30 |
| eHealth/e-learning websites, healthcare websites | 10 |
| General total | 1118 |

These results show that healthcare websites require continuous development and evaluation to achieve accessibility goals. Only through international cooperation following global standards can the WCAG-based website accessibility be improved, not only for disabled users but also for the public in general. The data and analyses from this study are accessible in the Mendeley Data [65] open repository for replication purposes.

Applying international standards like the WCAG facilitates an understanding and adaptation to current technological developments. This analysis ensures that health platforms can maintain accessibility over time and remain sustainable regarding equitable access to health information and services.

In addition, the availability of the data and analyses from this study in the Mendeley Data open repository promotes replication and transparency in research, contributing to the sustainable efforts to improve the accessibility of healthcare platforms worldwide.

Research question RQ10, which refers to the results obtained in the evaluation of the accessibility of health-related platforms, has a relationship with sustainability and is reflected in the advantage of using the WCAG and their updated versions (2.0, 2.1, 3.0) to evaluate the accessibility of health platforms. These guidelines are widely recognized and applied internationally, facilitating their understanding and adaptation to technological developments.

## 5. Discussion

The accessibility of healthcare platforms is essential, not only from an ethical point of view but also a functional one, ensuring that all people, regardless of their physical or

cognitive abilities, can access health information and services. We have obtained the series of conclusions and insights discussed below through this review.

The discussion of this study on web accessibility in the context of healthcare platforms allows us to explore the practical and theoretical implications of our findings. The main implications derived from our research highlight the importance of ensuring the accessibility of healthcare platforms for all users, regardless of their capabilities.

This review highlights that a combination of automated and manual tools appears to be the most comprehensive approach to assessing accessibility. While automatic tools, such as Achecker, WAVE, and TAW, provide a quick and systematic evaluation, a manual review gives a deeper and more contextual insight into the user experience. This duality is crucial because accessibility is about meeting specific technical guidelines and ensuring an optimal experience for the end user.

The results demonstrate that publications have followed updated versions of the WCAG over the years. While most websites have adopted the newer versions, some still use older versions. This finding underscores the need for developers and designers to stay updated with current guidelines as newer versions address emerging accessibility challenges.

Although a detailed list of specific errors has not been provided, there are recurring problems with the accessibility of health platforms. These failures can have serious consequences, especially in healthcare, where access to information and services can be vital. It is imperative that developers properly train in accessibility and conduct rigorous testing before releasing platforms.

The disabilities analyzed in the studies vary, raising the question of whether current assessments are comprehensive enough. Healthcare platforms should be accessible to everyone, including people with disabilities; as technology advances, so do the tools and techniques used to improve accessibility; therefore, it is crucial to stay informed and adapt to new developments.

The accessibility of health platforms is not an option but a necessity; although we have made progress in adopting tools and guidelines, there is still a long way to go to ensure everyone has equal access. Automated tools and manual assessments should be used together to provide a holistic picture of accessibility.

The literature review on the accessibility of healthcare platforms yielded significant findings and exciting figures that shed light on the current state of accessibility in this critical domain. These results can be a reference for future research endeavors to enhance the accessibility of healthcare platforms. Below, we will compare and discuss the critical findings of various authors selected in the literature review.

It was found that 52% of the articles used automated methods to assess accessibility, while 34% employed a combination of automatic and manual methods. These results align with the importance of using automated tools and manual reviews together, as emphasized by authors such as [15,38].

Popular tools like Achecker, WAVE, and TAW were widely utilized in accessibility evaluations, reflecting the trend identified by several authors in the literature, including [30,36].

Most articles adhered to the latest versions of the Web Content Accessibility Guidelines (WCAG); 70% of the studies focused on level A compliance. These findings are consistent with current recommendations and underscore the importance of following updated standards, as mentioned in several previous works [39].

The "Perceivable" category (WCAG Principle 1) was found to have the highest number of detected errors, with 39% of the articles mentioning issues in this area. This case aligns with the significance of adequately representing multimedia content and text alternatives, as authors such as [42] discussed.

The "Operable" and "Understandable" categories also exhibited significant errors, with 23% of the articles mentioning issues in each of these areas. These findings support the importance of navigability and content comprehension, as discussed in the literature [50].

Studies focused not only on disabilities but also on specific user groups, such as patients with chronic illnesses or palliative care needs. These instances highlight the importance of considering various user needs, as discussed in the works [45,56].

These results suggest critical areas for future research, such as developing more precise evaluation tools and focusing on specific user groups, such as patients with chronic illnesses.

Additionally, the need to continue updating and improving accessibility standards and ongoing training for designers and developers is emphasized, as suggested in the literature [41].

Implementing accessibility practices and technologies not only promotes equity in access to healthcare but can also improve the quality of care by enabling more effective communication and greater patient engagement.

Web accessibility is an ethical consideration and a legal [66] requirement in many countries. Our study underscores the importance of healthcare organizations complying with accessibility regulations [67], which can help them avoid legal issues and penalties. The legal implications of accessibility should be proactively considered.

Although implementing accessibility measures may require an initial investment, this study reveals that these investments can have a positive economic impact in the long term.

Improved accessibility can increase the user base and improve patient retention, translating into economic benefits for healthcare organizations. Web accessibility is relevant to healthcare organizations [68] and has broader societal implications. This research reveals that improvements in accessibility can contribute to a more inclusive society by ensuring that all people, regardless of their abilities, have equal access to healthcare information and services.

As a result of our research, the door is open for future research and developments in web accessibility in the healthcare context. For example, additional research could be conducted on emerging assistive technologies [14] and their impact on accessibility. In addition, user experiences [69] and their specific challenges could be further explored.

In the search for solutions to improve the accessibility of healthcare platforms, it is essential to consider the initiatives proposed by countries and the progress in assistive technologies in the healthcare sector.

Some countries have demonstrated a significant commitment to web accessibility in healthcare. For example, in Spain, the Ministry of Health [70] has produced a guide for web accessibility in the healthcare sector, providing valuable recommendations for creating accessible web pages and applications.

Similarly, in the United States, the Environmental Protection Agency [71] has developed tools and resources to support healthcare organizations in improving the accessibility of their websites. In addition, the UK government [72] has published an action plan that includes measures to improve web accessibility on government and public organization sites.

At the assistive technology level, leading companies such as Google [73], Microsoft [74], and Apple have played a crucial role in developing tools and resources that make it easier for people with disabilities to access the web. These initiatives include accessibility tools for their respective systems and platforms, demonstrating a commitment to online accessibility. In addition to these national-level initiatives, international organizations such as the World Wide Web Consortium (W3C) and the Web Accessibility Initiative (WAI) [10] have established the Web Content Accessibility Guidelines (WCAG) [4] as a set of essential recommendations for creating accessible web content.

We present some specific recommendations based on these initiatives to improve web accessibility in the healthcare sector. These recommendations include using clear and concise language, providing text alternatives for images, including captions on videos, offering formatting options for content, and including people with disabilities in testing. By following these recommendations, healthcare organizations can make a significant contribution to the accessibility of their platforms, ensuring they are accessible to all users.

This approach will help connect the initiatives proposed by countries and companies with the implications and recommendations of this study, highlighting how these initiatives can be relevant to improving accessibility in the healthcare sector.

The fact that some publications continue to use older versions of web accessibility guidelines highlights the importance of keeping up-to-date with current guidelines, as technology and accessibility needs evolve.

Finally, this discussion raises the importance of considering a wide range of disabilities and user needs in accessibility evaluations. It should be noted that this study focuses on sustainability because sustainable healthcare must be inclusive and accessible to everyone, regardless of their specific conditions or needs.

## 6. Conclusions, Limitations, and Future Work

This study has shed light on web accessibility in the context of healthcare platforms, highlighting the need for equitable access to online health information and services. As the digitization of healthcare platforms continues to expand, it becomes essential to rigorously address web accessibility challenges to ensure that all people, regardless of their disabilities or health conditions, can access and use these resources effectively.

This analysis has shown that common errors and considerations exist in designing healthcare platforms with respect to the WCAG, underscoring the importance of rigorous accessibility testing. The specific recommendations, such as using precise language, providing alternatives for multimedia content, and including formatting options, are valuable steps toward improving accessibility.

To move in this direction, healthcare organizations must consider including diverse user testing, focusing on different disabilities and health conditions. In addition, collaboration with assistive technologies and adopting tools and resources to improve accessibility are necessary steps.

Equity in access to health information and services is a fundamental goal that should guide the design and implementation of health platforms. As demonstrated in this study, web accessibility and sustainability are critical aspects of this goal. By addressing these challenges in a proactive and user-centered manner, healthcare organizations can contribute to a more inclusive and equitable online environment for all, consistent with the evidence and arguments presented in this paper.

This research sheds light on accessibility within healthcare platforms, revealing significant insights. The combined use of automated and manual evaluation methods emerged as a critical approach, with 52% of articles employing automated methods and 34% opting for a combined approach. Widely recognized tools such as Achecker, WAVE, and TAW are crucial in accessibility evaluations. In addition, 70% of the studies analyzed adhered to the most recent WCAG, focusing mainly on compliance with level A.

However, the "perceivable" category of the WCAG showed a considerable error rate, with 39% of articles highlighting problems in this area. In addition, some studies expanded their scope beyond disabilities to consider specific user groups, underscoring the need to address diverse user needs in healthcare platforms.

It is critical to acknowledge the limitations of this study; the scope was primarily limited to WCAG-related articles, so other relevant accessibility guidelines and approaches may have been overlooked. In addition, the analysis was based on 29 selected articles, which, although representative, may not comprehensively reflect the entire landscape of accessibility across healthcare platforms. The variability in accessibility evaluations may have introduced inconsistencies influenced by evaluation tools, scope, and evaluator experiences.

Future research efforts hold great promise for advancing the accessibility of healthcare platforms. The development of more sophisticated and accurate accessibility assessment tools, incorporating machine learning and artificial intelligence, has the potential to automate problem identification more effectively. Tailoring accessibility solutions to the needs and preferences of individual users, thereby improving their experience with healthcare platforms, represents an exciting avenue of exploration.

Adopting inclusive design principles beyond conventional accessibility guidelines can result in accessible healthcare interfaces. In addition, ongoing education and training programs for web designers, developers, and healthcare professionals can ensure that accessibility is prioritized. As the healthcare sector evolves technologically, research should contribute to adapting accessibility standards such as the WCAG to address new challenges.

The international collaboration between researchers, healthcare organizations, and accessibility experts can facilitate the sharing of best practices and resources to improve the accessibility of healthcare platforms around the world. Increasing reliance on telemedicine requires research attention to ensure equitable access, particularly in videoconferencing, remote monitoring, and for digital health records.

Exploring the legal and ethical dimensions of compliance with healthcare platform accessibility and ethical guidelines represents another vital avenue of research. In addition, longitudinal studies tracking progress in accessibility over time and regional variations in accessibility can provide valuable information. Assessing the role and effectiveness of emerging tools and techniques for improving platform accessibility and user-centered research can drive practical improvements.

This review study reveals that the combination of automated and manual evaluation methods has emerged as a critical approach, and this can be interpreted as a sustainable investment in the continuous improvement of accessibility.

Likewise, the need for ongoing education and training programs relates to sustainability in the sense that ensuring that there are accessibility experts in the future and that new professionals are prepared is essential to ensure the sustainability of the accessibility of healthcare platforms.

In future literature review work, we suggest incorporating assistive technologies, from the perspective of understanding how specialized hardware and software play a crucial role in ensuring that people with disabilities can effectively access online healthcare platforms by providing valuable solutions for improving web accessibility in the health context.

We suggest that future literature reviews address usability [69], accessibility, and sustainability in a way that includes opportunities for usability testing that looks at a wider diversity of users, addressing different disabilities and health conditions. In addition, we suggest, in future work, linking these values to an expanded assessment of a more significant number of platforms, and including emerging technologies in healthcare, which could be valuable research in the context of accessibility and sustainability in digital healthcare.

Finally, the call for international collaboration and the emphasis on adapting accessibility guidelines to new technological challenges indicate a long-term focus on the sustainability of accessibility in an ever-evolving healthcare environment.

**Author Contributions:** Conceptualization, P.A.-V., D.R.-S. and M.C.-T.; methodology, P.A.-V., D.R.-S., G.A.-V., M.C.-T., M.A.-C. and M.S.; validation, P.A.-V., D.R.-S., G.A.-V., M.A.-C. and M.G.-R.; formal analysis, P.A.-V., D.R.-S., G.A.-V., M.G.-R. and M.C.-T.; investigation, P.A.-V., D.R.-S., G.A.-V., M.S., M.G.-R., M.A.-C. and M.C.-T.; resources, P.A.-V.; writing—original draft preparation, P.A.-V., D.R.-S., G.A.-V., M.S., M.C.-T., E.O.-P., V.M.-G. and M.G.-R.; writing—review and editing, P.A.-V., D.R.-S., G.A.-V., M.S., M.C.-T., M.A.-C., E.O.-P., V.M.-G. and M.G.-R.; supervision, P.A.-V.; project administration, P.A.-V.; funding acquisition, P.A.-V. All authors have read and agreed to the published version of the manuscript.

**Funding:** This research was funded by the Corporación Ecuatoriana para el Desarrollo de la Investigación y Academia—CEDIA Project I+D+I-XVII-2022-27 and INI.PAV.23.01.

**Institutional Review Board Statement:** Not applicable.

**Informed Consent Statement:** Not applicable.

**Data Availability Statement:** Data are contained within the article.

**Acknowledgments:** The authors would like to thank the Corporación Ecuatoriana para el Desarrollo de la Investigación y Academia—CEDIA for their support of this research through the I+D+I-XVII-

2022-27 program, especially for the funding of the project: "Plataforma digital para educación terapéutica accesible hacia las necesidades de rehabilitación respiratoria".

**Conflicts of Interest:** The authors declare no conflict of interest.

## Appendix A

**Table A1.** Data extracted for research questions RQ1, RQ2, and RQ3.

| | **RQ1** | | | | **RQ2** | **RQ3** |
|---|---|---|---|---|---|---|
| **Paper ID** | **Type** | **Journal Name** | **Subject Area** | **Country** | **SJR** | **Pub, Year** |
| JiA22 | Journal | International Journal of Environmental Research and Public Health | Environmental Science | Switzerland | Q2 | 2022 |
| TeN22 | Journal | SN Computer Science | Computer Science | Germany | Q2 | 2022 |
| NaA22 | Journal | Universal Access in the Information Society | Computer Science | Germany | Q2 | 2022 |
| NaB22 | Conference | ACM SIGMM International Workshop | Computer Science | United States | N/A | 2022 |
| AnB22 | Conference | CEUR Workshop Proceedings | Computer Science | United States | N/A | 2022 |
| SeM22 | Journal | Universal Access in the Information Society | Computer Science | Germany | Q2 | 2022 |
| KuS22a | Conference | Smart Innovation, Systems and Technologies | Computer Science | Germany | Q4 | 2022 |
| KuS22b | Conference | Lecture Notes in Networks and Systems | Computer Science, Engineering | Switzerland | Q4 | 2022 |
| GlA21 | Conference | Lecture Notes in Networks and Systems | Computer Science, Engineering | Switzerland | Q4 | 2021 |
| PaA21 | Conference | Lecture Notes in Networks and Systems | Computer Science, Engineering | Switzerland | Q4 | 2021 |
| ElF20 | Journal | International Journal of Environmental Research and Public Health | Environmental Science, Medicine | Switzerland | Q2 | 2020 |
| MuA20 | Journal | Studies in health technology and informatics | Engineering, Medicine | Netherlands | Q3 | 2020 |
| YoJ20 | Journal | Universal Access in the Information Society | Computer Science | Germany | Q2 | 2020 |
| PaA20a | Conference | Advances in Intelligent Systems and Computing | Computer Science, Engineering | Germany | Q4 | 2020 |
| PaA20b | Conference | Communications in Computer and Information Science | Computer Science, Mathematics | Germany | Q4 | 2020 |
| PaA20c | Conference | Advances in Intelligent Systems and Computing | Computer Science, Engineering | Germany | Q4 | 2020 |
| PaA20d | Conference | Advances in Intelligent Systems and Computing | Computer Science, Engineering | Germany | Q4 | 2020 |
| CeS19 | Journal | Arabian Journal for Science and Engineering | Multidisciplinary | Germany | Q1 | 2019 |
| LuC19 | Journal | Gaceta Sanitaria | Medicine | Spain | Q3 | 2019 |
| PaA18a | Conference | IEEE Ecuador Technical Chapters Meeting (ETCM) | Computer Science, Engineering, | United States | N/A | 2018 |

**Table A1.** *Cont.*

| | RQ1 | | | | RQ2 | RQ3 |
|---|---|---|---|---|---|---|
| **Paper ID** | **Type** | **Journal Name** | **Subject Area** | **Country** | **SJR** | **Pub, Year** |
| PaA18b | Conference | International Conference on eDemocracy & eGovernment (ICEDEG) | Computer Science, Social Sciences | United States | N/A | 2018 |
| JoM17 | Journal | Computers in Human Behavior | Computer Science | United Kingdom | Q1 | 2017 |
| ArK17 | Conference | Confluence The Next Generation Information Technology Summit | Computer Science | United States | N/A | 2017 |
| EdL15 | Conference | Association of Information Systems (AIS) | Computer Science | United States | N/A | 2015 |
| LaO05 | Journal | Informatics for Health and Social Care | Health Professions, Medicine, Nursing, | United Kingdom | Q1 | 2005 |
| GrB22 | Journal | Disability and Health Journal | Medicine | United States | Q1 | 2022 |
| SaA21 | Journal | Journal of the American Medical Informatics Association | Medicine | United States | Q1 | 2021 |
| NoY18 | Journal | Business and Professional Communication Quarterly | Arts and Humanities, Business, Management, Economics | United States | Q2 | 2018 |
| NoY21 | Journal | Universal Access in the Information Society | Computer Science | Germany | Q2 | 2021 |

**Table A2.** Data extracted for research questions RQ4, RQ5, RQ6, and RQ7.

| | RQ4 | RQ5 | RQ6 | | | RQ7 |
|---|---|---|---|---|---|---|
| **Paper ID** | **Standards** | **Method Used** | **Automatic Tools** | **Real Users** | **Manual Tools** | **Disabilities** |
| JiA22 | WCAG 2.0/2.1 | Automatic and manual evaluation | Mauve++, Nibbler, WAVE, AChecker, SortSite | YES | Expert's review | Visual disability, auditory disability, physical disability, speech problem, cognitive/neurological problem, other disabilities (non-specified) |
| TeN22 | WCAG 2.0/2.1 | Automatic evaluation | Access Monitor, Achecker 2.1 | NO | N/A | Not specified (disabled users in general) |
| NaA22 | WCAG 2.0/2.1 Section 508 | Automatic evaluation | AChecker, WAVE, W3C HTML Validator, W3C CSS Validator | NO | N/A | Not specified (disabled users in general) |
| NaB22 | WCAG 2.0/2.1. ISO 9241-171:2008 [60] | Manual evaluation | Evaluation of Learning Management Systems (LMS): Moodle, Eliademy, Docebo, Sakai, Atutor | YES | Expert's review | Elderly users, children, people with chronic diseases, disabled people (visual, auditory, and motor impairments) |

**Table A2.** *Cont.*

| Paper ID | RQ4 | RQ5 | RQ6 | | | RQ7 |
|---|---|---|---|---|---|---|
| | Standards | Method Used | Automatic Tools | Real Users | Manual Tools | Disabilities |
| AnB22 | WCAG 2.0/2.1 | Manual evaluation | N/A | YES | Expert's review (from centers of reference, healthcare services, mobile units) | Poor people, disabled people, people with chronic conditions, older people, and children |
| SeM22 | WCAG 2.0 | Automatic evaluation | TAW, Deadlink Checker, Google Mobile-Friendly Test | NO | N/A | Not specified (disabled users in general) |
| KuS22a | WCAG 2.0 | Automatic evaluation | TAW, GTMetrix | NO | N/A | Not specified (disabled users in general) |
| KuS22b | WCAG 2.0 | Automatic evaluation | AChecker, WAVE, TAW | NO | N/A | Not specified (disabled users in general) |
| GlA21 | WCAG 2.1 | Automatic and manual evaluation | WAVE | NO | N/A | Not specified (disabled users in general) |
| PaA21 | WCAG 2.1 | Automatic evaluation | WAVE | NO | N/A | Not specified (disabled users in general) |
| ElF20 | WCAG 2.1 | Automatic and manual evaluation | WAVE | NO | N/A | Elderly users |
| MuA20 | WCAG 2.0 | Automatic evaluation, statistical analysis | Axe | NO | N/A | Elderly users |
| YoJ20 | WCAG 2.0 | Manual evaluation | N/A | YES | Real-users testing based on WCAG 2.0 | Visual disability: blind, second-level sight-impaired people |
| PaA20a | WCAG 2.1 | Automatic evaluation | WAVE | NO | N/A | Hip-arthroplasty patients |
| PaA20b | WCAG 2.0 | Automatic evaluation | WAVE | NO | N/A | Palliative care patients |
| PaA20c | WCAG 2.1 | Automatic and manual evaluation | WAVE, TAW | NO | N/A | Not specified (disabled users in general) |
| PaA20d | WCAG 2.1 | Automatic and manual evaluation | WAVE | NO | N/A | Arthroplasty patients |
| CeS19 | WCAG 2.0 | Automatic evaluation, statistical, quantitative, and qualitative analysis | Achecker, Nibbler | YES | User feedback questionnaires, expert's review | Users with accessibility issues: low-impaired vision, blind, motion problems, elderly, others |

**Table A2.** *Cont.*

| Paper ID | RQ4 Standards | RQ5 Method Used | RQ6 Automatic Tools | Real Users | Manual Tools | RQ7 Disabilities |
|---|---|---|---|---|---|---|
| LuC19 | WCAG 2.0 | Automatic evaluation | W3C XHTML Validator, CSS, eXaminator, TAW, Online diagnostic service of the WAO (Spanish government) | NO | N/A | Not specified (disabled and non-disabled users in general) |
| PaA18a | WCAG 2.0 | Automatic evaluation | WAVE, Siteimprove Accessibility Checker, OpenWAX, Tenon Check | NO | N/A | Arthroplasty patients, elderly users |
| PaA18b | WCAG 2.0 | Automatic and manual evaluation | Webometrics, WAVE, Tenon | YES | Expert's review | Disabled and elderly users |
| JoM17 | WCAG 2.0 | Automatic and manual evaluation | ACCESSWEB | YES | Real user experience, expert's review | Not specified (disabled and non-disabled users in general) |
| ArK17 | WCAG 2.0 | Automatic and manual evaluation | WebSiteOptimization, Readability-score.com, BuiltWith, TAW | YES | Expert's review (readability score and language analysis) | Not specified (disabled and non-disabled users in general) |
| EdL15 | WCAG 2.0 | Manual evaluation | NA | YES | Real users, expert's review, interviews, accessibility checklists, usability tests, questionnaires | Elderly users |
| LaO05 | WCAG 1.0 | Automatic evaluation | Bobby TM | NO | N/A | Not specified (disabled users in general) |
| GrB22 | WCAG 2.0 | Automatic evaluation, statistical analysis | WAVE | YES | Expert's review, accessibility rankings/score | Not specified (disabled users in general) |
| SaA21 | WCAG 2.0/2.1 | Automatic evaluation | AChecker, WAVE, SortSite | NO | N/A | Not specified (disabled users in general) |
| NoY18 | WCAG 2.0, Section 508 | Automatic evaluation | SortSite Professional Version 5.6 | NO | N/A | Elderly users |
| NoY21 | WCAG 2.0, Section 508 | Automatic and manual evaluation | AChecker | NO | N/A | Not specified (disabled users in general) |

**Table A3.** Data obtained for research questions RQ8, RQ9, and RQ10.

| | | RQ8 | RQ9 | | RQ10 | |
|---|---|---|---|---|---|---|
| **Paper ID** | **Standards** | **Conformance Levels** | **Describes Errors** | **Type of Errors** | **No. of Analyzed Websites** | **Sources** |
| JiA22 | WCAG 2.0/2.1 | A, AA, AAA | Yes | Related to Guidelines 1.1; 1.4; 2.4; 3.2 | 21 | COVID-19 vaccine information government websites from Europe and Asia |
| TeN22 | WCAG 2.0/2.1 | A, AA, AAA | Yes | Related to Guidelines 1.1; 1.4; 2.4 | 45 | COVID-19 information and vaccine registration public websites in Asian countries |
| NaA22 | WCAG 2.0/2.1, Section 508 | A, AA, AAA, | Yes | Success criteria 1.1.1; 1.3.1; 2.1.1; 2.4.2; 3.1.1; 3.3.2; 4.1.2 | 24 | Public health websites from North America, Europe, Asia, and South America |
| NaB22 | WCAG 2.0/2.1, ISO 9241-171:2008 [60] | A, AA, AAA | Yes | Errors not specified. Related to implementing an e-learning platform (i.e., outdated documentation, old operating systems) | 1 | Digital healthcare system: e-learning platform for increasing digital health literacy |
| AnB22 | WCAG 2.0/2.1 | A, AA, AAA | Yes | Errors not specified. Related to implementing an e-learning platform. | 1 | Digital healthcare system: e-learning platform for increasing digital health literacy |
| SeM22 | WCAG 2.0 | A, AA, AAA | Yes | Success criteria 1.1.1; 1.3.1; 2.4.4 | 58 | University hospital websites in Turkey (state and private) |
| KuS22a | WCAG 2.0 | A, AA, AAA | Yes | Related to principles 1, 3, and 4 | 10 | Five e-learning sites and five healthcare websites |
| KuS22b | WCAG 2.0 | AA | Yes | Related to principles 1, 2, 3, and 4 (most to least violated) | 6 | Top healthcare websites in India |
| GlA21 | WCAG 2.1 | A, AA | Yes | Related to principles 1 and 2 (alternative text, empty links, contrast errors). | 7 | Most visited healthcare websites in the world |
| PaA21 | WCAG 2.1 | AA | Yes | Related to principle 1 (alternative text, contrast errors). | 1 | Score Bebe website |
| ElF20 | WCAG 2.1 | A, AA, AAA | Yes | Success criteria 1.1.1; 1.3.1; 1.3.2; 2.1.1; 2.4.1; 2.4.2; 2.4.3; 2.4.4; 3.1.1; 3.3.1; 3.3.2; 4.1.1; 1.4.3; 1.4.4; 1.4.5; 2.4.6; 2.4.7; 3.2.3; 3.2.4; 3.3.3. 1.3.5; 1.4.11; 4.1.3 (Added in 2.1) | 6 | Representative pages from the entire WHO website |
| MuA20 | WCAG 2.0 | N/A | Yes | Related to principle 1. | 20 | Websites related to Alzheimer's from the UK and Indonesia |

**Table A3.** *Cont.*

| Paper ID | Standards | RQ8 Conformance Levels | RQ9 Describes Errors | Type of Errors | RQ10 No. of Analyzed Websites | Sources |
|---|---|---|---|---|---|---|
| YoJ20 | WCAG 2.0 | N/A | Yes | Related to principles 1, 2, 3, and 4 (i.e., alternate texts, link text, user responses, web accessibility). | 10 | Healthcare websites of the Korean government and public institutions |
| PaA20a | WCAG 2.1 | N/A | Yes | Related to principles 1 and 2 (i.e., low contrast, missing label alternative text for images). | 1 | ePHoRt project, web-based platform for hip arthroplasty patients' rehabilitation |
| PaA20b | WCAG 2.0 | A, AA, AAA | Yes | Principles 1, 2, and 3 Success criteria 1.1.1; 2.4.4 | 16 | Health-related websites, 11 from Ecuador and the top 5 websites according to Webometrics ranking |
| PaA20c | WCAG 2.1 | N/A | Yes | Principles 1, 2, 3, and 4 Success criteria 4.1.2; 1.1.1; 1.3.1 | 20 | Websites on health topics from the top places of Webometrics ranking |
| PaA20d | WCAG 2.1 | A, AA, AAA | Yes | Principles 1, 2, and 3 Success criteria 1.1.1; 1.3.1; 2.4.6; 2.4.4; 1.4.3; 3.3.2 | 6 | Telerehabilitation web pages extracted from the ePHoRt platform |
| CeS19 | WCAG 2.0 | A, AA, AAA | Yes | Principles 1 and 3 Success criteria 1.1.1; 1.3.1; 1.4.4; 1.4.6; 3.3.2 | 99 | Healthcare-related sites from nine European countries |
| LuC19 | WCAG 2.0 | A, AA, AAA | Yes | Related to principles 1 and 2 (no link to web map, missing headers, incorrect tags, others). | 18 | Websites related to online appointments for primary healthcare services in Spain |
| PaA18a | WCAG 2.0 | A, AA, AAA | Yes | Principles 1, 2, 3, and 4 Success criteria 3.1.1; 1.1.1; 2.4.4; 1.3.1; 2.4.6; 1.4.1; 4.1.1; 3.3.2; 2.4.7; 2.2.1; 2.4.1; 4.1.2; 1.4.3; 2.4.3; 2.4.9 | 1 | ePHoRt project, web-based platform for hip arthroplasty patients' rehabilitation |
| PaA18b | WCAG 2.0 | A, AA, AAA | Yes | Success criteria 1.1.1; 1.3.1; 1.3.2; 2.1.1; 2.4.1; 2.4.2; 2.4.3; 2.4.4; 2.4.6; 2.4.9; 2.4.10; 3.1.1; 3.3.2; 4.1.1; 4.1.2 | 22 | Hospital websites; 15 from the United States, one from France, two from Germany, two from Taiwan, one from the Netherlands, and one from Brazil (Webometrics ranking) |
| JoM17 | WCAG 2.0 | A, AA, AAA | Yes | Related to ALT text, TH elements, CSS property values, and others. | 20 | Iberian eHealth websites (Top 10 best and worst) |
| ArK17 | WCAG 2.0 | A, AA, AAA | Yes | Principles 1, 2, 3, and 4 Success criteria 1.1.1; 1.3.1; 2.4.4; 2.4.9; 2.4.10; 3.1.1; 3.3.2; 4.1.1; 4.1.2 | 379 | Hospital websites in Indian metro cities (Delhi, Mumbai, Kolkata, and Chennai) |

**Table A3.** *Cont.*

| | | RQ8 | RQ9 | | RQ10 | |
|---|---|---|---|---|---|---|
| **Paper ID** | **Standards** | **Conformance Levels** | **Describes Errors** | **Type of Errors** | **No. of Analyzed Websites** | **Sources** |
| EdL15 | WCAG 2.0 | N/A | Yes | Success criteria 1.1.1; 1.4.1; 1.4.4; 1.4.6; 1.4.8; 2.1.1; 2.4.2; 2.4.3; 2.4.4; 2.4.5; 2.4.7; 2.4.8; 3.2.1; 3.2.3; 3.3.2; 3.3.5 | 1 | Healthcare social network "My Health" research project from Fluminense Federal University Hospital |
| LaO05 | WCAG 1.0 | N/A | Yes | Priority 1 (missing ALT text) | 49 | Canadian consumer-oriented healthcare websites |
| GrB22 | WCAG 2.0 | N/A | Yes | Related to principles 1, 2, and 3 (insufficient text contrast, empty links, missing alt text, missing labels) | 56 | State/territory public health department COVID-19 information and vaccine web pages across the U.S. |
| SaA21 | WCAG 2.0/2.1 | A, AA, AAA | Yes | Related to principles 1, 2, 3, and 4 | 54 | COVID-19 vaccine registration websites in the U.S. |
| NoY18 | WCAG 2.0, Section 508 | A, AA | Yes | Section 508 errors related to Standard 1194.22 (A): Text for every non-text element shall be provided | 116 | VA Medical Center websites (U.S. Department of Veterans Affairs) |
| NoY21 | WCAG 2.0, Section 508 | A, AA, AAA | Yes | Success criteria 1.1.1; 1.3.1; 2.1.1; 2.4.1; 2.4.2; 2.4.4; 3.1.1; 3.2.2; 3.3.2; 4.1.1; 1.4.4; 2.4.6; 1.4.6 | 50 | U.S. State Occupational Safety and Health Agency (SOSHA) Consultation webpages |

**Table A4.** Principles, guidelines, and success criteria describing errors regarding WCAG 2.0 and WCAG 2.1.

| **Principles WCAG 2.0/2.1 [4,27]** [1] | **Level** | **Errors Described in Papers** |
|---|---|---|
| 1. Perceivable | | [15,36–40,43–45,50] [2] |
| Guideline 1.1 Text Alternatives: provide text alternatives for any non-text content so that it can be changed into other forms that people need. | | [30,31] [2] |
| 1.1.1 Non-text Content | A | [32,35,41,42,46–49,51,52,55,56] |
| Guideline 1.3 Adaptable: create content that can be presented in different ways (for example, a more straightforward layout) without losing information or structure. | | |
| 1.3.1 Info and Relationships | A | [32,35,41,42,47–49,51,52,55] |
| 1.3.2 Meaningful Sequence | A | [42,52] |
| 1.3.5 Identify Input Purpose | AA | [42] |
| Guideline 1.4 Distinguishable: make seeing and hearing content more accessible for users, including separating foreground from background. | | [30,31] [2] |
| 1.4.1 Use of Color | A | [51,56] |
| 1.4.3 Contrast (Minimum) | AA | [42,48,51] |
| 1.4.4 Resize Text | AA | [41,42,49,56] |

**Table A4.** *Cont.*

| Principles WCAG 2.0/2.1 [4,27] [1] | Level | Errors Described in Papers |
|---|---|---|
| 1.4.5 Images of Text | AA | [42] |
| 1.4.6 Contrast (Enhanced) | AAA | [41,49,56] |
| 1.4.8 Visual Presentation | AAA | [56] |
| 1.4.11 Non-text Contrast | AA | [42] |
| 2. Operable | | [15,37,38,40,44,45,50] [2] |
| Guideline 2.1 Keyboard Accessible: make all functionality available from a keyboard. | | |
| 2.1.1 Keyboard | A | [32,41,42,52,56] |
| Guideline 2.2 Enough Time: give users enough time to read and use content. | | |
| 2.2.1 Timing Adjustable | A | [51] |
| Guideline 2.4 Navigable: provide ways to help users navigate, find content, and determine where they are. | | [30,31] |
| 2.4.1 Bypass Blocks | A | [41,42,51,52] |
| 2.4.2 Page Titled | A | [32,41,42,52,56] |
| 2.4.3 Focus Order | A | [42,51,52,56] |
| 2.4.4 Link Purpose (In Context) | A | [35,41,42,46,48,51,52,55,56] |
| 2.4.5 Multiple Ways | AA | [56] |
| 2.4.6 Headings and Labels | AA | [41,42,48,51,52] |
| 2.4.7 Focus Visible | AA | [42,51,56] |
| 2.4.8 Location | AAA | [56] |
| 2.4.9 Link Purpose (Link Only) | AAA | [51,52,55] |
| 2.4.10 Section Headings | AAA | [52,55] |
| 3. Understandable | | [15,36,37,40,44,46,47] [2] |
| Guideline 3.1 Readable: make text content readable and understandable. | | |
| 3.1.1 Language of Page | A | [32,41,42,51,52,55] |
| Guideline 3.2 Predictable: make web pages appear and operate in predictable ways. | | [30] |
| 3.2.1 On Focus | A | [56] |
| 3.2.2 On Input | A | [41] |
| 3.2.3 Consistent Navigation | AA | [42,56] |
| 3.2.4 Consistent Identification | AA | [42] |
| Guideline 3.3 Input Assistance: help users avoid and correct mistakes. | | |
| 3.3.1 Error Identification | A | [42] |
| 3.3.2 Labels or Instructions | A | [32,41,42,48,49,51,52,55,56] |
| 3.3.3 Error Suggestion | AA | [42] |
| 3.3.5 Help | AAA | [56] |
| 4. Robust | | [15,36,40,44] [2] |
| Guideline 4.1 Compatible: maximize compatibility with current and future user agents, including assistive technologies. | | |
| 4.1.1 Parsing | A | [41,42,51,52,55] |
| 4.1.2 Name, Role, Value | A | [32,47,51,52,55] |
| 4.1.3 Status Messages | AA | [42] |

[1] Only the guidelines and success criteria in the analyzed papers are mentioned. [2] Papers with no specified errors but related to principles or guidelines.

## Appendix B

**Table A5.** Preferred Reporting Items for Systematic Reviews and Meta-Analyses Extension for Scoping Reviews (PRISMAScR) Checklist.

| Section | Item | PRISMA-Scr Checklist Item | Reported on Page # |
|---|---|---|---|
| | | Title | |
| Title | 1 | Identify the report as a scoping review. | 1 |
| | | Abstract | |
| Structured summary | 2 | Provide a structured summary that includes (as applicable) the background, objectives, eligibility criteria, sources of evidence, charting methods, results, and conclusions related to the review questions and objectives. | 1 |
| | | Introduction | |
| Rationale | 3 | Describe the rationale for the review in the context of what is already known. Please explain why the review questions/objectives lend themselves to a scoping review approach. | 1–3 |
| Objectives | 4 | Provide an explicit statement of the questions and objectives being addressed concerning their key elements (e.g., population or participants, concepts, and context) or other relevant critical elements used to conceptualize the review questions and/or objectives. | 3–5 |
| | | Methods | |
| Protocol and registration | 5 | Indicate whether a review protocol exists; state if and where it can be accessed (e.g., a web address); provide registration information, including the registration number if available. | 3–11 |
| Eligibility criteria | 6 | Specify the characteristics of the sources of evidence used as eligibility criteria (e.g., years considered, language, and publication status) and provide a rationale. | 3–11 |
| Information sources | 7 | Describe all information sources in the search (e.g., databases with dates of coverage and contact with authors to identify additional sources) and the date the most recent search was executed. | 3–11 |
| Search | 8 | Present the complete electronic search strategy for at least one database, including any limitations used so that it can be repeated. | 3–11 |
| Selection of sources of evidence | 9 | State the process for selecting sources of evidence (i.e., screening and eligibility) included in the scoping review. | 3–11 |
| Data charting process | 10 | Describe the methods of charting data from the included sources of evidence (e.g., calibrated forms or forms that the team has tested before their use and whether data charting was performed independently or in duplicate) and any processes for obtaining and confirming data from investigators. | 3–11 |
| Data items | 11 | List and define all variables for which data were sought and any assumptions and simplifications made. | 3–11 |
| Critical appraisal of individual sources of evidence | 12 | If done, provide a rationale for conducting a critical appraisal of the included sources of evidence; describe the methods used and how this information was used in any data synthesis (if appropriate). | 3–11 |
| Synthesis of results | 13 | Describe the methods of handling and summarizing the data that were charted. | 3–11 |
| | | Results | |
| Selection of sources of evidence | 14 | Give the number of sources of evidence screened, assessed for eligibility, and included in the review, with reasons for exclusions at each stage, ideally using a flow diagram. | 11–22 |
| Characteristics of sources of evidence | 15 | For each source of evidence, present characteristics for which data were charted and provide the citations. | 11–22 |
| Critical appraisal within sources of evidence | 16 | If carried out, present data on the critical appraisal of included sources of evidence (see item 12). | 11–22 |

<div align="center">Table A5. <em>Cont.</em></div>

| | | | |
|---|---|---|---|
| Results of individual sources of evidence | 17 | For each included source of evidence, present the relevant data that were charted that relate to the review questions and objectives. | 11–22 |
| Synthesis of results | 18 | Summarize and/or present the charting results related to the review questions and objectives. | 11–22 |
| Discussion | | | |
| Summary of evidence | 19 | Summarize the main results (including an overview of concepts, themes, and types of evidence available), link to the review questions and objectives, and consider the relevance to critical groups. | 21–25 |
| Limitations | 20 | Discuss the limitations of the scoping review process. | 21–25 |
| Conclusions | 21 | Provide a general interpretation of the results concerning the review questions, objectives, and potential implications and/or next steps. | 21–25 |
| Funding | | | |
| Funding | 22 | Describe funding sources for the included sources of evidence and the scoping review. Describe the role of the funders of the scoping review. | 25 |

PRISMA Extension for Scoping Reviews [22]. # Refers to the page number(s) on which the referenced topic is located.

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
