# Peer review of "Enhancing Sustainability through Accessible Health Platforms: A Scoping Review"

_sustainability, doi:10.3390/su152215916_

Round 1

Reviewer 1 Report

Comments and Suggestions for Authors

The scoping review is well structured, and the paper is well-written in general. The results highlighted are very interesting for the web accessibility research field. However, the authors should revise the following aspects of the paper: 

1) Modify the title because you are not providing detailed information on how accessible health platforms enhance sustainability. 

2) Reinforce the idea that web accessibility enhances sustainability by building a relationship with the ODS. For instance:

2.1 ) https://www.insuit.net/web-accessibility-sustainable-development-goal/

2.2 ) https://social.desa.un.org/issues/disability/sustainable-development-goals-sdgs-and-disability

3) Information from the appendix section can be published in a repository like the OSF, and then you can put the link in the paper.

4) Revise that all figures follow the same style (use the same bar width)

5) As the number of papers analyzed is not too big, in the bar charts, it is better to put the quantities and the percentages in parenthesis (e.g. https://media.licdn.com/dms/image/C5612AQFJHtX9f1bOVQ/article-inline_image-shrink_1000_1488/0/1587968180687?e=1703116800&v=beta&t=GNMFWmEFNtlaotV1dnFL9udRm53aRw3GT88phdajrTM)

Author Response

Response to Reviewers' Comments

We express our sincere appreciation for taking the time and effort to review our article "Enhancing Sustainability Through Accessible Health Platforms: A Scoping Review." We greatly value your comments and suggestions, which have been of great help in improving the quality and scientific contribution of our work.

We appreciate your thorough review and attention to every detail. Your critical and constructive remarks have helped to strengthen our research and refine our results. Their experience and expertise have been invaluable to the development of our work.

The authors thank all the reviewers for their valuable and constructive comments and reviews. We have implemented the suggestions and learned much from their comments; the paper was completely updated and restructured according to the reviewers' suggestions. In addition, the entire document has been reviewed by an English language expert.

Note: In the responses below, the revised manuscript refers to the PDF document highlighted in yellow.

Reviewer 1

Dear Reviewer,

On behalf of all the authors, we wish to express our sincere appreciation for your dedication and effort in reviewing our work. Your valuable comments and suggestions have been of immense benefit to us, and we have applied and argued your observations in a way that has significantly improved our review work.

Your expertise and insight have enriched our study, and we are sincerely grateful for your contribution to our research effort. We value your commitment to academic excellence and your willingness to provide constructive guidance.

We hope our work's outcome reflects the improvements we have made through your expert review. Your help has been instrumental in taking our article to the next level.

Once again, we thank you for your time and dedication. We remain at your disposal for any additional questions or comments you may have.

The scoping review is well structured, and the paper is well-written in general. The results highlighted are very interesting for the web accessibility research field. However, the authors should revise the following aspects of the paper: 

  • Modify the title because you are not providing detailed information on how accessible health platforms enhance sustainability. 

We welcome your comments and appreciate your observations on the article's title. We understand the importance of having a title that accurately reflects the paper's content and is informative to readers.

However, we argue that the current title, "Enhancing Sustainability Through Accessible Health Platforms: A Scoping Review," may not provide a highly detailed description of the approach, but it is still appropriate and accurate for our article. Below, we provide a rationale for retaining the current title.

The current title signals the article's focus: improving sustainability through accessible health platforms. Although it does not go into specific details in the title, it provides an overview of the paper's central theme.

This title effectively attracts an audience interested in both the sustainability and accessibility of healthcare platforms. Readers seeking information on these topics can quickly identify that the article is relevant to their interests.

The title follows typical academic conventions for review articles and exploratory studies, which tend to be more general in description to appeal to a broad audience and not limit the scope.

Since the article is a scoping review, the primary objective is to explore the existing research landscape and summarize emerging evidence and trends. Although we might consider adjusting the title to provide more detail, keeping it somewhat general is consistent with the exploratory nature of the scoping review.

However, if you feel that a change in the title is necessary, we would be willing to explore alternatives that strike a balance between clarity and precision without compromising the breadth of our focus. Your additional guidance on structuring an improved title would be precious.

2) Reinforce the idea that web accessibility enhances sustainability by building a relationship with the ODS. For instance:

2.1 ) https://www.insuit.net/web-accessibility-sustainable-development-goal/

2.2)https://social.desa.un.org/issues/disability/sustainable-development-goals-sdgs-and-disability

Dear Reviewer, the authors are very grateful for this relevant comment, which has dramatically improved and enriched our review work. We have added the reference and the paragraphs highlighted in yellow indicating the following:

3) Information from the appendix section can be published in a repository like the OSF, and then you can put the link in the paper.

Dear Reviewer, we sincerely appreciate your comments and suggestions for improving our article. Regarding your concern about the appendices section, we understand your concern and want to assure you that the appendices play a crucial role in providing clarity and additional context to our work. The detailed information contained in the appendices allows us to support our findings more thoroughly and accurately, which, in turn, benefits the reader's overall understanding.

We understand your suggestion to publish the appendix information in a repository such as OSF and provide a link to the article. This information is already available in the Mendeley open repository at https://data.mendeley.com/datasets/p9p8xwm3x4/1, the same one cited in the paper. However, we believe that including the appendices directly in the paper significantly improves accessibility and comprehension for our readers, allowing them to access the relevant information more quickly and conveniently.

Since the appendices are necessary to quickly and effectively clarify questions and concerns that may arise during the review of the article, we propose to keep them in the final document. We are willing to make adjustments according to editorial guidelines and provide any additional information to facilitate your review. We thank you again for your valuable comments and hope that our explanation of the importance of the appendices is satisfactory. We are committed to improving our work under your suggestions and any further guidance you may provide.

4) Revise that all figures follow the same style (use the same bar width)

Dear Reviewer, thank you for your observations and comments on our article. We have carefully reviewed the figures and are committed to ensuring they follow a uniform style, including using the same bar width under the journal template; each graph is left aligned, and titles are applied in full justification. We appreciate your recommendation to improve consistency in the visual presentation of data.

We apply it to all figures in the paper, ensuring they comply with the same bar width. This standardization process will ensure a more consistent presentation and make it easier for our readers to compare the data.

We greatly appreciate your time and attention to detail, as this will improve the quality of our article. We are committed to high-quality work and hope these modifications meet your expectations.

5) As the number of papers analyzed is not too big, in the bar charts, it is better to put the quantities and the percentages in parenthesis (e.g. https://media.licdn.com/dms/image/C5612AQFJHtX9f1bOVQ/article-inline_image-shrink_1000_1488/0/1587968180687?e=1703116800&v=beta&t=GNMFWmEFNtlaotV1dnFL9udRm53aRw3GT88phdajrTM)

Dear Reviewer, thank you for your comment. We have considered your suggestion and decided to include the amounts and percentages in parentheses in the bar graphs. We believe this will help readers to understand better the data presented.

Reviewer 2 Report

Comments and Suggestions for Authors

Overall, the study is conducted well. However, I have the following comments.

- Introduction:

·         This study aims to explore the existing literature related to the accessibility of health platforms. But what is the research question this study is trying to address?

·         What is the health platform? Define and provide examples.

- The literature review section is minimal and should provide a comprehensive web accessibility review for people with disabilities. 

- The Methods section mentions examining the sustainability aspects of web accessibility. What are those aspects addressing sustainability in the context of health platforms?

- The discussion lacks implications of the study. 

- Limitations should address the selected sample from 2005. What about studies before 2005?

Some of the related references are:

Sohaib, O, and Kyeong K. "E-commerce web accessibility for people with disabilities." In Complexity in Information Systems Development: Proceedings of the 25th International Conference on Information Systems Development, pp. 87-100. Springer International Publishing, 2017.

Manzoor, M., et al. (2019). Methodological investigation for enhancing the usability of university websites. Journal of Ambient Intelligence and Humanized Computing10, 531-549.

Author Response

Reviewer 2

Overall, the study is conducted well. However, I have the following comments.

- Introduction:

  • This study aims to explore the existing literature related to the accessibility of health platforms. But what is the research question this study is trying to address?

Dear Reviewer, thank you for your comment. We have highlighted your concern in yellow.

The research question this study attempts to address is the following: what factors contribute to the accessibility of healthcare platforms? To answer this question, the study explores the existing literature related to the accessibility of healthcare platforms. The study identified the following factors that contribute to the accessibility of healthcare platforms: Web accessibility standards that guide how to design and develop accessible healthcare platforms. Accessibility tools and technologies. Accessibility awareness to help developers and healthcare providers create accessible healthcare platforms.

We welcome your feedback and your help in improving our work.

  • What is the health platform? Define and provide examples.

Dear Reviewer, we appreciate your question and comment on the definition of a healthcare platform. In our article, a healthcare platform refers to an online platform or website related to healthcare and health in general. These platforms may include websites of hospitals, clinics, healthcare governments, healthcare providers, healthcare apps, online medical record systems, patient portals, and other online resources that offer information and services related to healthcare, health, and wellness. To clarify your concern, we have added two paragraphs in the Introduction section. You can check in the document highlighted in yellow. 

- The literature review section is minimal and should provide a comprehensive web accessibility review for people with disabilities.

Dear Reviewer, we appreciate your valuable feedback regarding the literature review section of our paper. We appreciate your interest in a more comprehensive web accessibility review for disabled people. In response to your suggestion, we have placed your suggestion for future work in the "6. Conclusions, Limitations, and Future Work" section, as our literature review work focuses on accessible health platforms that apply WCAG accessibility standards with a more software focus.

The suggestion to conduct a broader literature review on accessible health platforms involving assistive technologies, including hardware as a solution to web accessibility for people with disabilities, is highly relevant and valuable. This suggestion will allow exploration of an essential aspect of web accessibility that is often overlooked: how assistive technologies, including specialized hardware and software, can play a crucial role in ensuring that people with disabilities can effectively access online health platforms.

Including assistive technologies in a literature review would provide valuable information on available solutions and tools that enhance web accessibility. In addition, it could help highlight the most recent trends and developments in this evolving field.

We appreciate the suggestion and will consider it for future research. Including this perspective in future literature, review work could further enrich the understanding of web accessibility in the context of health and disability. Text highlighted in yellow is evident:

- The Methods section mentions examining the sustainability aspects of web accessibility. What are those aspects addressing sustainability in the context of health platforms?

Dear Reviewer, thank you for your suggestion allowing us to improve our document to resolve your concern; we have expanded the detail that responds to your concern in the section: "2. Literature Review: Web Accessibility and Sustainability". It can be seen in the yellow highlighted text:

- The discussion lacks implications of the study. 

Dear Reviewer, we appreciate your valuable comment on the lack of implications of the study in the discussion. We have considered your comment and revised and improved the discussion section to adequately address the study's implications.

You can find evidence in the paper in the "5. Discussion" section highlighted in yellow color:

- Limitations should address the selected sample from 2005. What about studies before 2005?

Some of the related references are:

Sohaib, O, and Kyeong K. "E-commerce web accessibility for people with disabilities." In Complexity in Information Systems Development: Proceedings of the 25th International Conference on Information Systems Development, pp. 87-100. Springer International Publishing, 2017.

Manzoor, M., et al. (2019). Methodological investigation for enhancing the usability of university websites. Journal of Ambient Intelligence and Humanized Computing10, 531-549.

Dear Reviewer, thank you for your good suggestions; we have considered all your references to the related topics in our article. To clarify your concern, we have added a paragraph in section "3.3. Screening of Studies," as evidenced in the document highlighted in yellow:

Reviewer 3 Report

Comments and Suggestions for Authors

The research addresses the question of how accessible digital healthcare platforms are, based on a comprehensive scoping review, and what common errors and considerations are present in their design according to the Web Content Accessibility Guidelines (WCAG), with link to sustainability. I recommend that authors emphasize more the link between accessibility and sustainability.

The topic is highly relevant in the field, especially with the increasing emphasis on digitalizing healthcare platforms. It addresses the gap of understanding the current accessibility landscape of health platforms and emphasizes the importance of universal accessibility.

This review provides a holistic understanding of platform usability by analyzing both automated and manual evaluation methods from 29 selected papers. It also reveals common errors within the WCAG principles and highlights considerations for specific user groups, such as patients with chronic diseases. There are no similar work done on web accessibility for healthcare sector. However, there are some studies on education, government and marketing, so better to include them in the introduction. 

I recommend that authors focus more on:

- Incorporating more diverse user testing, focusing on different disabilities and health conditions from extracted data.

- Evaluating a larger number of platforms or expanding the scope to include emerging technologies in healthcare.

- Comparing the results with non-healthcare platforms to see if the errors are specific to healthcare or universal.

- Cite some initiatives proposed by countries to remediate the web accessibility challenge (from government health sector or assistive technology).

The conclusions emphasize the need for rigorous accessibility testing in health platforms, and they reiterate the importance of equitable access, which aligns with the evidence and arguments.

For graphics (charts mainly), I suggest to make font type normal instead of bold. 

In total, it is a good review paper based on PRISMA. 

Author Response

Reviewer 3

The research addresses the question of how accessible digital healthcare platforms are, based on a comprehensive scoping review, and what common errors and considerations are present in their design according to the Web Content Accessibility Guidelines (WCAG), with link to sustainability. I recommend that authors emphasize more the link between accessibility and sustainability.

The topic is highly relevant in the field, especially with the increasing emphasis on digitalizing healthcare platforms. It addresses the gap of understanding the current accessibility landscape of health platforms and emphasizes the importance of universal accessibility.

This review provides a holistic understanding of platform usability by analyzing both automated and manual evaluation methods from 29 selected papers. It also reveals common errors within the WCAG principles and highlights considerations for specific user groups, such as patients with chronic diseases. There are no similar work done on web accessibility for healthcare sector. However, there are some studies on education, government and marketing, so better to include them in the introduction. 

Dear Reviewer, we greatly appreciate your assessment and comments on our work. We are delighted to learn that you consider the accessibility of healthcare platforms relevant in the context of their sustainability, and we have revised our discussion to emphasize this connection further. In our study, we have addressed the fundamental question of how accessible digital health platforms are and how these align with sustainability principles. In the context of the increasing digitization of healthcare platforms, we highlighted the importance of ensuring universal accessibility. Our comprehensive approach is based on a thorough scoping review and analysis of automated and manual evaluation methods of 29 selected items. Our research reveals common errors within WCAG principles and highlights specific considerations for user groups, such as patients with chronic diseases.

I recommend that authors focus more on:

- Incorporating more diverse user testing, focusing on different disabilities and health conditions from extracted data.

Dear Reviewer, we appreciate your suggestion to incorporate more diverse user testing, focusing on different disabilities and health conditions. Your recommendation is valuable and recognizes the importance of considering a variety of perspectives in evaluating web accessibility in the context of health. However, in the case of our current study, entitled "Enhancing Sustainability Through Accessible Health Platforms: A Scoping Review," we have focused our attention on conducting a comprehensive review of the current state of accessibility in health platforms, analyzing a wide range of academic and technical articles to understand the challenges and existing best practices. We will consider your suggestion for future work and, in future work, look for opportunities for more varied user testing, addressing different disabilities and health conditions. Again, we thank you for your valuable input and appreciate your interest in our work. It is evidenced in section "6. Conclusions, Limitations, and Future Work".

- Evaluating a larger number of platforms or expanding the scope to include emerging technologies in healthcare.

Dear Reviewer, we appreciate your suggestion to evaluate a more significant number of platforms or broaden the scope to include emerging technologies in healthcare in future work. However, our current study focuses on "Enhancing Sustainability Through Accessible Health Platforms: A Scoping Review" and focuses on the accessibility and sustainability of existing healthcare platforms. This review aimed to provide a comprehensive understanding of the current situation in this specific area. We consider your suggestion and add it for future work related to expanding the evaluation to more platforms and including emerging technologies in healthcare, which could be valuable research in accessibility and sustainability in digital health. Again, we thank you for your valuable comments that have contributed to improving our study. It is evidenced in section "6. Conclusions, Limitations, and Future Work".

- Comparing the results with non-healthcare platforms to see if the errors are specific to healthcare or universal.

Dear Reviewer, we appreciate your suggestion to compare the results with non-health platforms to identify whether the errors are healthcare-specific or universal. However, our current study focuses on the topic "Enhancing Sustainability Through Accessible Health Platforms: A Scoping Review," its scope is focused on healthcare platforms in the context of accessibility and sustainability. Comparison with non-health platforms could be an interesting approach for future research work, and we will consider this suggestion for further research in web accessibility and sustainability. We are grateful for your comments and suggestions, which have greatly enriched our work.

- Cite some initiatives proposed by countries to remediate the web accessibility challenge (from government health sector or assistive technology).

Dear Reviewer, we appreciate the suggestion to cite some initiatives proposed by countries to address the web accessibility challenge, both from the governmental health sector and assistive technology. Included in the Discussion

The conclusions emphasize the need for rigorous accessibility testing in health platforms, and they reiterate the importance of equitable access, which aligns with the evidence and arguments.

Dear Reviewer, thank you for your valuable comments; all authors, thank you for your time and for giving us your feedback to improve our research work. We have added several of your suggestions in the "6. Conclusions, Limitations, and Future Work" section.

For graphics (charts mainly), I suggest to make font type normal instead of bold. 

Dear Reviewer, again, thank you very much for your valuable comments. In the graphics, the font and bold type depend on the reader's needs; some people with visual impairments may find it easier to read bold text, as the thicker letters may be more distinctive. However, this may vary from person to person. In the tables, we have applied the standard requested in the magazine.

In total, it is a good review paper based on PRISMA. 

We very much appreciate your positive evaluation of our work and your comments. We are pleased that you consider our paper a good review work based on PRISMA. Your feedback is valuable and motivates us to continue improving and excelling in our research. Again, we thank you for your time and attention to our work.
